# A Theory of Multi-Agent Generative Flow Networks

## Abstract

Generative flow networks utilize a flow-matching loss to learn a stochastic policy for generating objects from a sequence of actions, such that the probability of generating a pattern can be proportional to the corresponding given reward. However, a theoretical framework for multi-agent generative flow networks (MA-GFlowNets) has not yet been proposed. In this paper, we propose the theory framework of MA-GFlowNets, which can be applied to multiple agents to generate objects collaboratively through a series of joint actions. We further propose four algorithms: a centralized flow network for centralized training of MA-GFlowNets, an independent flow network for decentralized execution, a joint flow network for achieving centralized training with decentralized execution, and its updated conditional version. Joint Flow training is based on a local-global principle allowing to train a collection of (local) GFN as a unique (global) GFN. This principle provides a loss of reasonable complexity and allows to leverage usual results on GFN to provide theoretical guarantees that the independent policies generate samples with probability proportional to the reward function. Experimental results demonstrate the superiority of the proposed framework compared to reinforcement learning and MCMC-based methods.

## 1 Introduction

Generative flow networks (GFlowNets) Bengio et al. (2023) can sample a diverse set of candidates in an active learning setting, where the training objective is to approximate sampling of the candidates proportionally to a given reward function. Compared to reinforcement learning (RL), where the learned policy is more inclined to sample action sequences with higher rewards, GFlowNets can perform exploration tasks better. The goal of GFlowNets is not to generate a single highest-reward action sequence, but rather is to sample a sequence of actions from the leading modes of the reward function Bengio et al. (2021). However, based on current theoretical results, GFlowNets cannot support multi-agent systems.

A multi-agent system is a set of autonomous interacting entities that share a typical environment, perceive through sensors, and act in conjunction with actuators Busoniu et al. (2008). Multi-agent reinforcement learning (MARL), especially cooperative MARL, is widely used in robot teams, distributed control, resource management, data mining, etc Zhang et al. (2021); Canese et al. (2021); Feriani & Hossain (2021). There two major challenges for cooperative MARL: scalability and partial observability Yang et al. (2019); Spaan (2012). Since the joint state-action space grows exponentially with the number of agents, coupled with the environment's partial observability and communication constraints, each agent needs to make individual decisions based on the local action observation history with guaranteed performance Sunehag et al. (2018); Wang et al. (2020); Rashid et al. (2018). In MARL, to address these challenges, a popular centralized training with decentralized execution (CTDE) paradigm Oliehoek et al. (2008); Oliehoek & Amato (2016) is proposed, in which the agent's policy is trained in a centralized manner by accessing global information and executed in a decentralized manner based only on the local history. However, extending these techniques to GFlowNets is not straightforward, especially in constructing CTDE-architecture flow networks and finding IGM conditions for flow networks need investigating.

In this paper, we propose the multi-agent generative flow networks (MA-GFlowNets) framework for cooperative decision-making tasks. Our framework can generate more diverse patterns through

sequential joint actions with probabilities proportional to the reward function. Unlike vanilla GFlowNets, the proposed method analyzes the interaction of multiple agent actions and shows how to sample actions from multi-flow functions. Our approach consists of building a virtual global GFN capturing the policies of all agents and ensuring consistency of local (agent) policies. Variations of this approach yield different flow-matching losses and training algorithms.

Furthermore, we propose the Centralized Flow Network (CFN), Independent Flow Network (IFN), Joint Flow Network (JFN), and Conditioned Joint Flow Network (CJFN) algorithms for multi-agent GFlowNets framework. CFN considers multi-agent dynamics as a whole for policy optimization regardless of the combinatorial complexity and demand for independent execution, so it is slower; while IFN is faster, but suffers from the flow non-stationary problem. In contrast, JFN and CJFN, which are trained based on the local-global principle, takes full advantage of CFN and IFN. They can reduce the complexity of flow estimation and support decentralized execution, which are beneficial to solving practical cooperative decision-making problems.

**Main Contributions:** 1) We first generalize the measure GFlowNets framework to the multi-agent setting, and propose a theory of multi-agent generative flow networks for cooperative decision-making tasks; 2) We propose four algorithms under the measure framework, namely CFN, IFN, JFN and CJFN, for training multi-agent GFlowNets, which are respectively based on centralized training, independent execution, and the latter two algorithms are based on the CTDE paradigm; 3) We propose a local-global principle and then prove that the joint state-action flow function can be decomposed into the product form of multiple independent flows, and that a unique Markovian flow can be trained based on the flow matching condition; 4) We conduct experiments based on cooperative control tasks to demonstrate that the proposed algorithms can outperform current cooperative MARL algorithms, especially in terms of exploration capabilities.

## 2    PROBLEM FORMULATION

The **multi-agent setting** formalizes the data of state, actions and transitions for multiple agents. The state space $\mathcal{S}$ as well as the state-dependent action spaces $\mathcal{A}_s$ are measurable spaces; for each state $s \in \mathcal{S}$, the environment comes with a stochastic transition map[1] $\mathcal{A}_s \xrightarrow{T_s} \mathcal{S}$. We formalize this dependency on state by bundling (packing) state and action together into a bundle $\{(s,a) \mid s \in \mathcal{S}, a \in \mathcal{A}_s\} = \mathcal{A} \xrightarrow{S,T} \mathcal{S}$ where $S(s,a) := s$ and $T(s,a) := T_s(a)$. For graphs, a bundled action is an edge $s \to s'$, the statemap $S$ returns the origin $s$ while the transition map returns the destination $s'$. A policy is then a section of $S$ ie a kernel $\mathcal{S} \xrightarrow{\pi} \mathcal{A}$ such that $S \circ \pi$ is identity on $\mathcal{S}$.

Each agent $i \in I$ in the finite agent set $I$ has its own observation $o^{(i)}$ in its observation space $\mathcal{O}^{(i)}$; it depends on the state via the projection $\mathcal{S} \xrightarrow{p^{(i)}} \mathcal{O}^{(i)}$. For simplicity sake, we identify $\mathcal{S} = \prod_{i \in I} \mathcal{O}^{(i)}$. Each agent has its own action space $\mathcal{A}^{(i)}$ and each of the agent observation-dependent action space $\mathcal{A}_o$ contains a special action STOP; the environment is such that once an agent chooses STOP, it is put on hold until all agents do as well. The game finishes when all agent have chosen STOP, a reward is given based on the last state. The reward received is formalized by a non-negative function $r : \mathcal{S} \to \mathcal{S}$. We assume that each agent may freely choose its own action

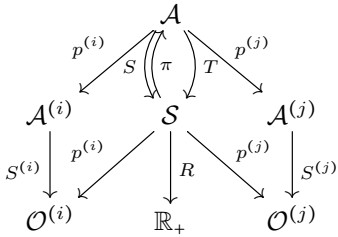

Figure 1: Multi-agent formalism

independently from the actions chosen by other agents: this is formalized via $\mathcal{A}_s = \prod_{i \in I} \mathcal{A}^{(i)}_{o^{(i)}} / \sim$ ie the Cartesian product of agent actions space up to identification of the STOP actions. A trajectory of the system of agents is a, possibly infinite, sequence of states $(s_t)_{t < \tau+1}$ with $\tau \in \mathbb{N} \cup \{\infty\}$ starting at the source state $s_0 \in \mathcal{S}$ and may eventually calling STOP; the space of trajectories is $\mathcal{T}$. A policy on $\mathcal{S}$ induces a Markov chain hence a distribution on trajectories.

*Our objective is to build a policy $\pi$ so that the induced trajectories are finite and $s_\tau$ is distributed proportionally to $R := r\lambda$ where $\lambda$ is some fixed measure on $\mathcal{S}$ and $\int_{s \in \mathcal{S}} r(s)d\lambda(s)$ is finite.*

---

[1]We adopt the naming convention of Douc et al. (2018). The kernel $K : \mathcal{X} \to \mathcal{Y}$ is a stochastic map which is formalized as follows: for all $x \in \mathcal{X}$, $K(x \to \cdot)$ is a probability distribution on $\mathcal{Y}$. In addition, $K(x \to \cdot)$ varies measurably with $x$ in the sense that for all measurable set $A \subset \mathcal{Y}$, the real valued map $x \mapsto K(x \to A)$ is measurable.

**Measurable GFlowNets** Brunswic et al. (2024); Lahlou et al. (2023); Li et al. (2023d); Deleu & Bengio (2023); Bengio et al. (2023) are defined in the single-agent setting i.e.

$$\mathcal{A} \overset{S}{\underset{T}{\overleftarrow{\pi}}} \mathcal{S} \xrightarrow{R} \mathbb{R}_+ \ , \ \text{with } |I| = 1.$$

A GFlowNets on $(\mathcal{S}, \mathcal{A}, S, T, R)$ is a forward policy $\pi : \mathcal{S} \to \mathcal{A}$ together with a non-negative finite measure $F_{\text{out}}$ on $\mathcal{S}$ called the outflow or state-flow. The reward is generally non-trainable and unknown but implicitly a component of $F_{\text{out}}$ and $\pi$; since the reward may not be tractable in the multi-agent setting, we favor a reward-free parameterization of GFlowNets. We thus parameterize them by triplets $(\pi^*, F_{\text{out}}^*, F_{\text{init}})$ where $\pi^*(s) = \pi(s \mid a \neq \text{STOP})$, $F_{\text{out}}^* := F_{\text{out}} - R$ and $F_{\text{init}} = F_{\text{out}}(s_0)\pi(s_0)$. The $*$-notations informally mean we restrict the objects to $\mathcal{S} \smallsetminus \{s_0, s_f\}$. Given a GFlowNet in $*$-notations together with a reward, there is a unique GFlowNet in usual notations.

A GFlowNet is trained to satisfy the so-called flow-matching constraint:

$$F_{\text{out}} = F_{\text{in}} := F_{\text{init}} + F_{\text{out}}^* \pi^* T, \tag{1}$$

as measures on $\mathcal{S}$. In passing we introduce $\hat{R} := F_{\text{in}} - F_{\text{out}}^*$, $F_{\text{in}}^* := F_{\text{out}}^* \pi^* T$ and $F_{\text{action}} := F_{\text{out}} \otimes \pi$. The induced Markov chain starts at $s_0$ sampled from the unnormalized distribution $F_{\text{init}}$ and then for every $t$ the policy is applied until the action STOP is picked: $a_t \sim \pi(s_t \to \cdot)$ and if $a_t \neq \text{STOP}$, $s_{t+1} \sim T(a_t \to \cdot)$. Usually we choose $F_{\text{init}} \propto \ell(\theta)$ with $\ell$ a known, easily sampled from, distribution family. The sampling time $\tau$ is the $t$ such that $a_t = \text{STOP}$.

The following Theorem was first proved on DAG in Bengio et al. (2023) and shows GFlowNets answer our problem definition.

**Theorem 1 ((Brunswic et al., 2024) Theorem 2)** *Let* $\mathbb{F} := (\pi, F_{\text{out}}^*, F_{\text{init}})$ *be a GFlowNets on* $(\mathcal{S}, \mathcal{A}, S, T, R)$. *If the reward* $R$ *is non-zero and* $\mathbb{F}$ *satisfies the flow-matching constraint, then its sampling time is almost surely finite and the sampling distribution is proportional to* $R$. *More precisely:*

$$\mathbb{P}(\tau < +\infty) = 1, \qquad \mathbb{E}(\tau) \leq \frac{F_{\text{out}}(\mathcal{S})}{R(\mathcal{S})} - 1, \quad \text{and} \quad s_\tau \sim \frac{1}{R(\mathcal{S})}R. \tag{2}$$

**Flow-matching losses (FM)**, denoted by $\mathcal{L}_{\text{FM}}$, compare the outflow $F_{\text{out}}$ with the inflow $F_{\text{in}} := F_{\text{init}} + F_{\text{out}}T\pi$; They are minimized when $F_{\text{in}} = F_{\text{out}}$ so that, surely, a gradient descent on GFlowNets parameters enforces equation 1. In the original works Bengio et al. (2021); Malkin et al. (2022), Bengio et al. used divergence-based FM losses valid as long as the state space does not have cycle and Brunswic et al. (2024) introduced stable FM losses allowing training in presence of cycles:

$$\mathcal{L}_{\text{FM}}^{\text{div}}(\mathbb{F}^\theta) = \mathbb{E}_{s \sim \nu_{\text{state}}} g \circ \log\left(\frac{dF_{\text{in}}^\theta}{dF_{\text{out}}^\theta}(s)\right) \tag{3}$$

$$\mathcal{L}_{\text{FM}}^{\text{stable}}(\mathbb{F}^\theta) = \mathbb{E}_{s \sim \nu_{\text{state}}} g\left(\frac{dF_{\text{in}}^\theta}{d\lambda}(s) - \frac{dF_{\text{out}}^\theta}{d\lambda}(s)\right), \tag{4}$$

where $g$ is some positive function, decreasing on $[-\infty, 0]$, $g(0) = 0$ and increasing on $[0, +\infty]$. A practical stable training loss on graphs can be written as

$$\mathcal{L}(\mathbb{F}^\theta) = \mathbb{E} \sum_{t=1}^{\tau} \left\{ \log\left[1 + \varepsilon \left|F_{\text{in}}^\theta(s_t) - F_{\text{out}}^\theta(s_t)\right|^\alpha\right] \times \left(1 + \eta\left(F_{\text{in}}^\theta(s_t) + F_{\text{out}}^\theta(s_t)\right)\right)^\beta \right\}, \tag{5}$$

where $s_t$ are path sampled from **any** distribution of paths in $\mathcal{S}$, and the parameters satisfy the condition $\{\varepsilon, \eta, \alpha, \beta > 0\}$.

**MA-GFlowNets** are tuples $((\mathbb{F}^{(i)})_{i \in I}, \mathbb{F})$, where each *local* GFlowNets $\mathbb{F}^{(i)}$ is defined on $(\mathcal{O}^{(i)}, \mathcal{A}^{(i)}, S^{(i)}, T^{(i)}, R^{(i)})$ for $i \in I$ and the *global* GFlowNets $\mathbb{F}$ is defined on $(\mathcal{S}, \mathcal{A}, S, T, R)$. In general, some GFlowNets (local or global) may be virtual in the sense that it is not implemented.

# 3 MULTI-AGENT GFLOWNETS

This section is devoted to details and theory regarding the variations of algorithms for MA-GFlowNets training. If resources allow, the most direct approach is included in the training of the global model directly, leading to a centralized training algorithm in which the local GFlowNets are virtual. As expected, such an algorithm suffers from high computational complexity, hence, demanding decentralized algorithms. Decentralized algorithms require the agents to collaborate to some extent. We achieve such a collaboration by enforcing consistency rules between the local and global GFlowNets. The global GFlowNets is virtual and is used to build a training loss for the local models ensuring the global model is GFlowNets, so that the sampling Theorem applies. The sampling properties of the MA-GFlowNets are then deduced from the flow-matching property of the virtual global model.

## 3.1 CENTRALIZED TRAINING

Centralized training consists in training of the global flow directly. Here, the local flows are virtual in the sense that they are recovered from the global flow as image by the observation maps. We use FM-losses as given in equations 3-4 applied to the flow on $(\mathcal{S}, \mathcal{A})$. See Algorithm 1. Implicitly, $F_{\text{out}}$ contains a parameterizable component from $F_{\text{out}}^*$, while $F_{\text{in}}$ contains the parameterization of $\pi^*$ and $F_{\text{init}}$.

---

**Algorithm 1** Centralized Flow Network Training Algorithm for MA-GFlowNets

**Input:** A multi-agent environment $(\mathcal{S}, \mathcal{A}, \mathcal{O}^{(i)}, \mathcal{A}^{(i)}, p_i, S, T, R)$, a parameterized GFlowNets $\mathbb{F} :=$ $(\pi, F_{\text{out}}^*, F_{\text{init}})$ on $(\mathcal{S}, \mathcal{A})$.
    **while** not converged **do**
        Sample and add trajectories $(s_t)_{t \geq 0} \in \mathcal{T}$ to replay buffer with policy $\pi(s_t \to a_t)$.
        Generate training distribution $\nu_{\text{state}}$.
        Apply minimization step of the FM loss $\mathcal{L}_{\text{FM}}^{\text{stable}}(\mathbb{F}^\theta)$ .
    **end while**

---

From the algorithmic viewpoint, the CFN algorithm is identical to a single GFlowNets. As a consequence, usual results on the measurable GFlowNets apply as is. There are, however, a number of key difficulties: 1) even on graphs, the computational complexity increases as $O(|\mathcal{A}_s|^N)$ at any given explored state; 2) centralized training requires all agents to share observations, which is impractical since in many applications the agents only have access to their own observations.

## 3.2 LOCAL TRAINING: INDEPENDENT

The dual training method is embodied in the training of local GFlowNets instead of the global one. In this case, the local flows $\mathbb{F}^{(i)}$ are parameterized and the global flow is virtual. In the same way, a local FM loss is used for each client. In order to have well-defined local GFlowNets, we need a local reward, for which a natural definition is $R^{(i)}(o_t^{(i)}) := \mathbb{E}(R(s_t)|o_t^{(i)})$. The local training loss function can be written as:

$$\mathcal{L}(\mathbb{F}^{(i)}) = \mathbb{E} \sum_{t=1}^{\tau} \left\{ \log \left[ 1 + \varepsilon \left| F_{\text{in}}^{\theta_i}\left(o_t^i\right) - F_{\text{out}}^{\theta_i}\left(o_t^i\right) \right|^{\alpha} \right] \times \left( 1 + \eta \left( F_{\text{in}}^{\theta_i}\left(o_t^i\right) + F_{\text{out}}^{\theta_i}\left(o_t^i\right) \right) \right)^{\beta} \right\}. \quad (6)$$

The algorithm 3 in Appendix B describes a simplest training method, which solves the issue of exponential action complexity with an increasing number of agents. In this formulation, however, two issues arise: the evaluation of ingoing flow $F_{\text{in}}^{(i)}(o^{(i)})$ becomes harder as we need to find all transitions leading to a given local observation (and not to a given global state). This problem may be non-trivial as it is also related to the actions of other agents. More importantly, in this case, the local reward is intractable so we cannot accurately estimate the reward $R^{(i)}(o^{(i)})$ of each node; Falling back to using the

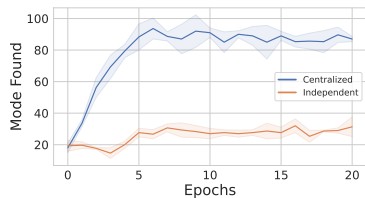

Figure 2: Performance comparison on Hyper-grid task.

stochastic reward $R^{(i)}(o^{(i)}) := R(s_t|o_t^{(i)})$ instead leads to transition uncertainty and spurious rewards, which can cause non-stationarity and/or mode collapse as shown in Figure 2.

### 3.3 LOCAL-GLOBAL TRAINING

#### 3.3.1 LOCAL-GLOBAL PRINCIPLE: JOINT FLOW NETWORK

Local-global training is based upon the following local global principle which combined with Theorem 1 ensures that the MA-GFlowNet have sampling distribution proportional to the reward $R$.

**Theorem 2 (Joint MA-GFlowNets)** *Given local GFlowNets $\mathbb{F}^{(i)}$ on some environments $(\mathcal{O}^{(i)}, \mathcal{A}^{(i)}, S^{(i)}, T^{(i)})$ there exists a global GFlowNets $\mathbb{F}^{\mathrm{joint}}$ on a multi-agent environment $(\prod_{i \in I} \mathcal{O}^{(i)}, \mathcal{A}, S, \tilde{T})$ consistent with the local GFlowNets $\mathbb{F}^{(i)}$, such that*

$$F_{\mathrm{out}}^* = \prod_{i \in I} F_{\mathrm{out}}^{(i),*}, \quad F_{\mathrm{in}} = \prod_{i \in I} F_{\mathrm{in}}^{(i)}. \tag{7}$$

*Moreover, if $\mathbb{F}^{\mathrm{joint}}$ satisfies equation 1 for a reward $R$ and each $\hat{R}^{(i)} \geq 0$ then $R = \prod_{i \in I} \hat{R}^{(i)}$.*

Our Joint Flow Network (JFN) algorithm, leverage Theorem 2 by sampling trajectories with policy

$$o_t^{(i)} = p_i(s_t^{(i)}) \text{ and } \pi^{(i)}(o_t^{(i)} \to a_t^{(i)}), \ \ i \in I \tag{8}$$

with $a_t = (a_t^{(i)} : i \in I)$ and $s_{t+1} = T(s_t, a_t)$, build formally the (global) joint GFlowNet from local GFlowNets and train the collection of agent via the FM-loss of the joint GFlowNet. Equation 7 ensures that the inflow and outflow of the (global) joint GFlowNet are both easily computable from the local inflows and outflows provided by agents. See algorithm 2.

---

**Algorithm 2** Joint Flow Network Training Algorithm for MA-GFlowNets

**Input:** Number of agents $N$, A multi-agent environment $(\mathcal{S}, \mathcal{A}, \mathcal{O}^{(i)}, \mathcal{A}^{(i)}, p_i, S, T, R)$.
**Input:** Local parameterized GFlowNets $(\pi^{(i),*}, F_{\mathrm{out}}^{(i),*}, F_{\mathrm{init}}^{(i)})_{i \in I}$.
 **while** not converged **do**
  Sample and add trajectories $(s_t)_{t \geq 0} \in \mathcal{T}$ to replay buffer with policy according to equation 8.
  Generate training distribution of states $\nu_{\mathrm{state}}$ from the replay buffer.
  Apply minimization step of the FM loss $\mathcal{L}_{\mathrm{FM}}^{\mathrm{stable}}(\mathbb{F}^{\theta,\mathrm{joint}})$ for reward $R$.
 **end while**

---

This training regiment presents two key advantages: over centralized training, the action complexity is linear w.r.t. the number of agents and local actions as in the independent training; over independent training, the reward is not spurious. Indeed, in $\mathcal{L}_{\mathrm{FM}}^{\mathrm{stable}}(\mathbb{F}^{\theta,\mathrm{joint}})$, by equation 7, the computation of $F_{\mathrm{in}}$ and $F_{\mathrm{out}}^*$ reduces to computing the inflow and star-outflow for each local GFlowNets. Also, only the global reward $R$ appears. The remaining, possibly difficult, challenge is the estimation of local ingoing flows from the local observations as it depends on the local transitions $T^{(i)}$, see first point below. At this stage, the relations between the global/joint/local flow-matching constraints are unclear, and furthermore, the induced policy of the local GFlowNets still depends on the yet undefined local rewards. The following point clarify those links.

First, the collection of local GFlowNets induces local transitions kernels $T^{(i)} : \mathcal{O}^{(i)} \to \mathcal{O}^{(i)}$ which are not uniquely determined in general by a single GFlowNets. Indeed, the local policies induce a global policy $\pi(s_t \to a_t) := \prod_{i \in I} \pi(o_t^{(i)} \to a_t^{(i)})$. Then, the (virtual) transition kernel $T^{(i)}(a_t^{(i)}) = (T(a_t)|a_t^{(i)})$ of the GFlowNets $i$ depends on the distribution of states and the corresponding actions of **all** local GFlowNets. See appendix A.5 for details. Note that $T^{(i)}$ are derived from the actual environment $T$ and the joint GFlowNets on the multi-agent environment with the true transition $T$, while the Theorem above ensures splitting of star-inflows and virtual rewards only for the approximated $\tilde{T}$. Furthermore, local rewards may be formalized as a stochastic reward to take into account the lack of information of a single agent, but they are never used during training: the allocation of rewards across agents is irrelevant. Only the virtual rewards $\hat{R}^{(i)} = F_{\mathrm{out}}^{(i),*} - F_{\mathrm{in}}^{(i)}$ are relevant but they are effectively free. As a consequence, Algorithm 2 effectively trains both the

joint flow as well as a product environment model. But since in general $T \neq \tilde{T}$ Algorithm 2 may fail to reach satisfactory convergence.

Second, beware that in our construction of the joint MA-GFlowNets, there is no guarantee that the global initial flow is split as the product of the local initial flows. In fact, we favor a construction in which $F_{\text{init}}$ is non-trivial to account for the inability of local agents to assess synchronization with another agent. See Appendix A.8 for formalization details.

Third, we may partially link local and global flow-matching properties.

**Theorem 3** *Let $(\mathbb{F}^{(i)})_{i \in I}$ be local GFlowNets and let $\mathbb{F}$ be their joint GFlowNets. Assume that none of the local GFlowNets are zero and that each $\hat{R}^{(i)} \geq 0$. If $\mathbb{F}$ satisfies equation 1, then there exists an "essential" subdomain of each $\mathcal{O}^{(i)}$ on which local GFlowNets satisfy the flow-matching constraint.*

The restriction regarding the domain on which local GFlowNets satisfy the flow-matching constraint is detailed in Appendix A.8, this sophistication arises because of the stopping condition of the multi-agent system. The essential domain may be informally formulated as "where the local agent is still playing": an agent may decide (or be forced) to stop playing, letting other agents continue playing, the forfeited player is then on hold until the game stops and rewards are actually awarded.

To conclude, the joint GFlowNets provides an approximation of the target global GFlowNets, this approximation may fail if the transition kernel $T$ is highly coupled or if the reward is not a product.

### 3.3.2 CONDITIONED JOINT FLOW NETWORK

As discussed training of MA-GFlowNets via training of the virtual joint GFlowNets is an approximation of the centralized training. In fact, the space of joint GFlowNets is smaller than that of the general MA-GFlowNets, as only rewards that splits into the product $R(s) = \prod_{i \in I} R^{(i)}(o^{(i)})$ may be exactly sampled. If the rewards are not of this form, the training may still be subject to a spurious reward or mode collapse. For instance, consider the case of $\mathcal{S} = \{1, 2\}^2$ with two agents of respective positions $s_1, s_2 \in \{1, 2\}$, actions $\{(0, +1), (+1, 0), (0, 0), (+1, +1)\}$, and reward $R(s_1, s_2) = \mathbf{1}_{s_1 == s_2}$. In this case, the reward does not split and it is easy to see that independent agents cannot sample states proportionally to $R$. One may easily build more sophisticated counter-examples based on this one.

Our proposed solution is to build a conditioned JFN inspired by augmented flows Dupont et al. (2019); Huang et al. (2020) methods, which allow the bypass of architectural constraints for Normalization flows Papamakarios et al. (2021). The trick is to add a shared "hidden" state to the joint MA-GFlowNets allowing the agent to synchronize. This hidden state is constant across a given episode and may be understood as a cooperative strategy chosen beforehand by the agents. The size of this hidden parameterization is a tradeoff: it should be large enough to allow the proper parameterization of the target reward and transition but the larger the size the harder the training. Formally, this simply consist in augmenting the state space and the observation spaces by a strategy space $\Omega$ to get $\tilde{\mathcal{S}} = \mathcal{S} \times \Omega$ and $\tilde{\mathcal{O}}^{(i)} = \mathcal{O}^{(i)} \times \Omega$, $F_{\text{init}}$ is augmented by a distribution $\mathbb{P}$ on $\Omega$, the observation projections as well as transition kernel act trivially on $\Omega$ ie $T(s; \omega) = T(s)$ and $p^{(i)}(s; \omega) = (p^{(i)}(s), \omega)$. The joint MA-GFlowNets theorem applies the same way, beware that the observation part of $T^{(i)}$ now have a dependency on $\Omega$ even though $T$ does not. In theory, $\Omega$ may be big enough to parameterize the whole trajectory space $\mathcal{T}$, in which case it is possible to have decoupled conditioned local transition kernels $T^{(i)}(\cdot; \omega)$ so that $\tilde{T} = T$ on a relevant domain. Furthermore, the limitation on the reward is also lifted if the flow-matching property is enforced on the expected joint flow $\mathbb{E}_\omega \mathbb{F}^{\text{joint}}$. Two possible losses may be considered: $\mathbb{E}_\omega \mathcal{L}_{\text{FM}}^{\text{stable}}(\mathbb{F}^{\theta, \text{joint}}(\cdot; \omega))$ or $\mathcal{L}_{\text{FM}}^{\text{stable}}(\mathbb{E}_\omega \mathbb{F}^{\theta, \text{joint}}(\cdot; \omega))$. The former, which we use in our experiments, is simpler to implement but does not a priori lift the constraint on the reward.

The training phase of Conditioned Joint Flow Network (CJFN) is shown in Algorithm 4 in the appendix. We first sample trajectories with policy

$$o_t^{(i)} = p_i(s_t^{(i)}) \text{ and } \pi_\omega^{(i)}(o_t^{(i)} \to a_t^{(i)}), \quad i \in I \tag{9}$$

with $a_t = (a_t^{(i)} : i \in I)$ and $s_{t+1} = T(s_t, a_t)$. Then we train the sampling policy by minimizing the FM loss $\mathbb{E}_\omega \mathcal{L}_{\text{FM}}^{\text{stable}}(\mathbb{F}^{\theta, \text{joint}}(\cdot; \omega))$.

**Discussion:** Finally, we discuss the connection between MA-GFlowNets and multi-agent RL in Appendix C and prove some related properties.

# 4 RELATED WORKS

**Generative Flow Networks:** GFlowNets is an emerging generative model that could learn a policy to generate the objects with a probability proportional to a given reward function. Nowadays, GFlowNets has achieved promising performance in many fields, such as molecule generation Bengio et al. (2021); Malkin et al. (2022); Jain et al. (2022), discrete probabilistic modeling Zhang et al. (2022), structure learning Deleu et al. (2022), domain adaptation Zhu et al. (2023), graph neural networks training Li et al. (2023b;a), and large language model training Li et al. (2023c); Hu et al. (2023); Zhang et al. (2024). This network could sample the distribution of trajectories with high rewards and can be useful in tasks where the reward distribution is more diverse.

GFlowNets is similar to reinforcement learning (RL) Sutton & Barto (2018). However, RL aims to maximize the expected reward and often only generates the single action sequence with the highest reward. Conversely, the learned policies of GFlowNets can ensure that the sampled actions are proportional to the reward, making them more suitable for exploration. This exploration ability makes GFlowNet promising as a new paradigm for policy optimization in the RL field, but there are many problems, such as solving multi-agent collaborative tasks. Previously, the meta GFlowNets algorithm Ji et al. (2024) was proposed to solve the problem of GFlowNets training under distributed conditions, but it requires the observation state and task objectives of each agent to be the same, which is not suitable for multi-agent problems. Later, a multi-agent GFlowNets algorithm was proposed in Luo et al. (2024), but this algorithm is an approximate algorithm without theoretical support and is difficult to converge when solving large-scale multi-agent problems. In contrast, we established the theory of multi-agent GFlowNets in measure space, and our algorithm can support large-scale multi-agent environments, such as StarCraft missions.

**Cooperative Multi-agent Reinforcement Learning:** There exist many MARL algorithms to solve collaborative tasks. Two extreme algorithms for thus purpose are independent learning Tan (1993) and centralized training. Independent training methods regard the influence of other agents as part of the environment, but the team reward function often has difficulty to measure the contribution of each agent, resulting in the agent facing a non-stationary environment Sunehag et al. (2018); Yang et al. (2020).

On the contrary, centralized training treats the multi-agent problem as a single-agent counterpart. However, this method has high combinatorial complexity and is difficult to scale beyond dozens of agents Yang et al. (2019). Therefore, the most popular paradigm is centralized training and decentralized execution (CTDE), including value-based Sunehag et al. (2018); Rashid et al. (2018); Son et al. (2019); Wang et al. (2020) and policy-based Lowe et al. (2017); Yu et al. (2022); Kuba et al. (2022) methods. The goal of value-based methods is to decompose the joint value function among the agents for decentralized execution. This requires satisfying the condition that the local maximum of each agent's value function should be equal to the global maximum of the joint value function. The methods, VDN Sunehag et al. (2018) and QMIX Rashid et al. (2018) employ two classic and efficient factorization structures, additivity and monotonicity, respectively, despite their strict factorization method.

In QTRAN Son et al. (2019) and QPLEX Wang et al. (2020), extra design features are introduced for decomposition, such as the factorization structure and advantage function. The policy-based methods extend the single-agent TRPO Schulman et al. (2015) and PPO Schulman et al. (2017) into the multi-agent setting, such as MAPPO Yu et al. (2022), which has shown surprising effectiveness in cooperative multi-agent games. The goal of these algorithms is to find the policy that maximizes the long-term reward. However, it is difficult for them to learn more diverse policies in order to generate more promising results.

# 5 EXPERIMENTS

We first verify the performance of CFN on a multi-agent hyper-grid domain where partition functions can be accurately computed. We then compare the performance of CFN and CJFN with stan-

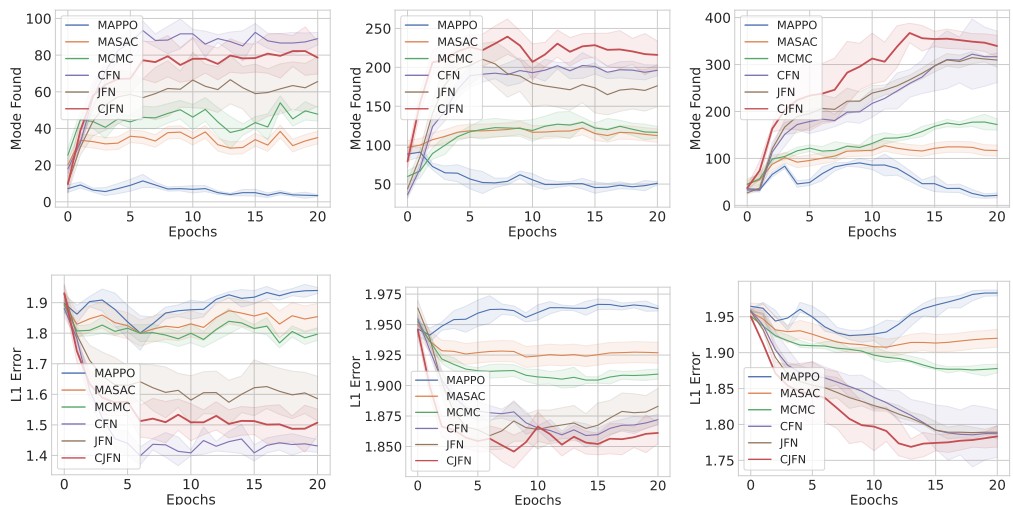

Figure 3: Mode Found and L1 error performance of different algorithms on various hyper-grid environments. Top and bottom are respectively Mode Found (higher is better) and L1 Error (lower is better). **Left:** Hyper-Grid v1, **Middle:** Hyper-Grid v2, **Right:** Hyper-Grid v3.

dard MCMC and some RL methods to show that our proposed sampling distributions better match normalized rewards. All our code is done using the PyTorch Paszke et al. (2019) library. We re-implemented the multi-agent RL algorithms and other baselines.

## 5.1 HYPER-GRID ENVIRONMENT

We consider a multi-agent MDP where states are the cells of a $N$-dimensional hypercubic grid of side length $H$. In this environment, all agents start from the initialization point $x = (0, 0, \cdots)$, and are only allowed to increase coordinate $i$ with action $a_i$. In addition, each agent has a stop action. When all agents choose the stop action or reach the maximum $H$ of the episode length, the entire system resets for the next round of sampling. The reward function is designed as

$$R(x) = R_0 + R_1 \prod_i \mathbb{I}\left(0.25 < |x_i/H - 0.5|\right) + R_2 \prod_i \mathbb{I}\left(0.3 < |x_i/H - 0.5| < 0.4\right), \quad (10)$$

where $x = [x_1, \cdots, x_I]$ includes all agent states and the reward term $0 < R_0 \ll R_1 < R_2$ leads a distribution of modes.

By changing $R_0$ and setting it closer to $0$, this environment becomes harder to solve, creating an unexplored region of state space due to the sparse reward setting. We conducted experiments in Hyper-grid environments with different numbers of agents and different dimensions. We used different version numbers to differentiate these environments, where the higher the number is, the more the number of dimensions and proxies are. The specific details about the environments and experiments can be found in the appendix.

We compare CFN and CJFN with a modified MCMC and RL methods. In the modified MCMC method Xie et al. (2021), we allow iterative reduction of coordinates on the basis of joint action space and cancel the setting of stop actions to form a ergodic chain. As for the RL methods, we consider the maximum entropy algorithm, i.e., multi-agent SAC Haarnoja et al. (2018), and a previous cooperative multi-agent algorithm, i.e., MAPPO, Yu et al. (2022). Note that the maximum entropy method uses the Softmax policy of the value function to make decision, so as to explore the state of other reward, which is related to our proposed algorithm. To measure the performance of these methods, we use the normalized L1 error as $\mathbb{E}[|p(s_f) - \pi(s_f)| \times N]$ with $p(s_f) = R(s_f)/Z$ being the sample distribution computed by the true reward, where $N$ is cardinality of the space of $s_f$. Moreover, we can consider the mode found theme to demonstrate the superiority of the proposed algorithm.

Figure 3 illustrates the performance superiority of our proposed algorithm compared to other methods in the L1 error and Mode Found. We find that on small-scale environments shown in Figure 3-Left, CFN can achieve the best performance because CFN can accurately estimate the flow of joint actions when the joint action space dimension is small. There

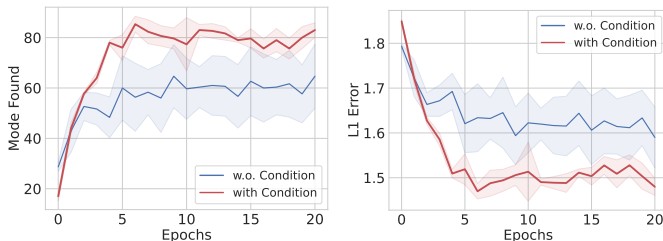

Figure 4: Comparison results of JFN and Conditional JFN.

are two main reasons for the large l1-error index. First, we normalized the standard L1-error and multiplied it by a constant to avoid the inconvenience of visualization of smaller magnitude. Secondly, when evaluating L1-error, we only sampled 20 rounds for calculation, and with the increase of the number of samples, L1-error will further decrease. As the complexity of the joint action flow that needs to be estimated increases, we find that the performance of CFN degrades. However, the joint-flow based methods still achieve good estimation and maintain the speed of convergence, as shown in Figure 3-Middle. Note that the RL-based methods do not achieve the expected performance. Their performance curves first rise and then fall, because as training progresses, these methods tend to find the highest rewarding nodes rather than finding more patterns. Figure 4 shows the performance superiority of the CJFN. When the algorithm introduces conditions to coordinate multiple agents, the performance is closer to the optimal.

## 5.2 STARCRAFT

Figure 5 shows the performance of the proposed algorithm on the StarCraft 3m map, where (a) shows the win rate comparison with different algorithms, and (b) and (c) show the decision results sampled using the proposed algorithm. In the experiment, the outflow flow is calculated when the flow function is large, and the maximum flow is used to calculate the win rate when sampling. It can be found that since the experimental environment is not a sampling environment with diversified rewards, although the proposed algorithm is not significantly better than other algorithms, it still illustrates its potential in large-scale decision-making. In addition, the proposed algorithm can sample results with more diverse rewards, such as (b) and (c), and the number of units left implies the trajectory reward. More detailed results are given in the Appendix. One thing to note is that the task of the benchmark is to achieve as high a win rate as possible, which is somewhat different from the goal of GFLowNets, but it can be used to verify the effectiveness of the algorithm.

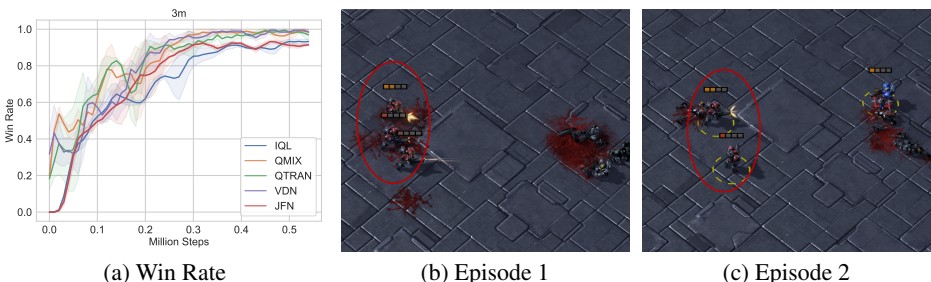

(a) Win Rate      (b) Episode 1      (c) Episode 2

Figure 5: The performance comparison results on the 3m map of StarCraft

## 6 CONCLUSION

In this paper, we discussed the policy optimization problem when GFlowNets meets the multi-agent systems. Different from RL, the goal of MA-GFlowNets is to find diverse samples with probability proportional to the reward function. Since the joint flow is equivalent to the product of independent flow of each agent, we designed a CTDE method to avoid the flow estimation complexity prob-

lem in a fully centralized algorithm and the non-stationary environment in the independent learning process, simultaneously. Experimental results on Hyper-Grid environments and StarCraft task demonstrated the superiority of the proposed algorithms.

**Limitation and Future Work:** Our theory is incomplete as it does not apply to non-cooperative agents and has limited support of different game/agent terminations or initialization. A local-global principle beyond independent agent policies would also be particularly interesting. Our experiments do not cover the whole range of the theory in particular regarding continuous tasks and CJFN loss on projected GFN. An ablation study analyzing the tradeoff of small versus big condition space $\Omega$ would enlighten its importance. Finally, a metrization of the space of global GFlowNet would allow a more precise functional and optimization analysis of JFN/CJFN and their limitations.

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

# A  JOINT FLOW THEORY

The goal of this section is to lay down so elementary points on the measurable theory of MA-GFlowNets as well as prove the main theorem on the joint GFlowNet.

## A.1  NOTATIONS ON MEASURES AND KERNELS

We mostly use notations from Douc et al. (2018) regarding kernels and measures. In the whole section, since we deal with technicalities, we use kernel type notations for image by kernels and maps (seen as deterministic kernels). So that for a kernel $K : X \to Y$ and a measure $\mu$ on $X$ we denote by $\mu K$ the measure on $Y$ defined by $\mu K(B) = \int_{x \in X} K(x \to B) d\mu(x)$ for $B \subset Y$ measurable and $\mu \otimes K$ is the measure on $X \times Y$ so that $\mu \otimes K(A \times B) = \int_{x \in A} K(x \to B) d\mu(x)$. Recall that a measure $\nu$ dominates a measure $\mu$ which is denoted $\mu \ll \nu$, if for all measurable $A$, $\nu(A) = 0 \Rightarrow \mu(A) = 0$. The Radon-Nykodim Theorem ensures that if $\mu \ll \nu$ and $\mu, \nu$ are finite then there exists $\varphi \in L^1(\nu)$ so that $\mu = \varphi \nu$. This function $\varphi$ is called the Radon-Nykodim derivative and is denoted $\frac{d\mu}{d\nu}$. We favor notations $\mu(A \to B)$ when $\mu$ is a measure on $X \times Y$ and $A \subset X$ and $B \subset Y$; also $\mu(A \to \cdot)$ means the measure $B \mapsto \mu(A \to B)$.

## A.2  AN INTRODUCTION FOR NOTATIONS

We understand that our formalism is abstract, this section is devoted justifying our choices and providing examples.

### A.2.1  MOTIVATIONS

To begin with, our motivation to formalize the action space as a measurable bundle $\mathcal{A} := \{(s, a) \mid s \in \mathcal{S}, a \in \mathcal{A}_s\} \xrightarrow{S} \mathcal{S}$ is three fold:

1. The available actions from a state may depend on the state itself: on a grid, the actions available while on the boundary of the grid are certainly more limited than while in the middle. More generally, on a graph, actions are typically formalized by edges $s \xrightarrow{a} s'$ of the graph, the data of an edge contains both the origin $s$ and the destination $s'$. In other words, on graphs, actions are bundled with an origin state. It is thus natural to consider the actions as bundled with the origin state. When an agent is transiting from a state to another via an action, the state map tells where it comes from while the transition map tells where it is going.

2. We want our formalism to cover as many cases as possible in a unified way: Graphs, vector spaces with linear group actions or mixture of discrete and continuous state spaces. Measures and measurable spaces provide a nice formalism to capture the quantity of reward on a given set or a probability distribution.

3. We want a well-founded and possibly standardized mathematical formalism. In particular, the policy takes as input a state and returns a distribution of actions. the actions should correspond to the input state. To avoid having a cumbersome notion of policy as a family of distributions $\pi_s$ each on $\mathcal{A}_s$, we prefer to consider the union of the state-dependent action spaces $\mathcal{A} := \bigcup_{s \in \mathcal{S}} \mathcal{A}_s$ and define the policy as Markov kernel $\mathcal{S} \to \mathcal{A}$. However, we still need to satisfy the constraint that the distribution $\pi(s)$ is supported by $\mathcal{A}_s$. Bundles are usual mathemcatical objects formalizing such situations and constraints and are thus well suited for this purpose and the constraint is easily expressed as $S \circ \pi(s) = s, \forall s \in \mathcal{S}$.

Our synthetic formalism comes with a few drawbacks due to the level of abstraction:

1. The notation $\pi(s)$ differs from the more common notation $\pi(s, a)$ as the action already contains $s$ implicitly.

2. We need to use Radon-Nikodym derivative. At a given state, on a graph, a GFlowNets has a probability to stop

$$\mathbb{P}(\text{STOP}|s) = \frac{R(s)}{F_{\text{out}}(s)}.$$

On a continuous statespace with reference measure $\lambda$, the stop probability is

$$\mathbb{P}(\text{STOP}|s) = \frac{r(s)}{f_{\text{out}}(s)}$$

where $r(s)$ is the density of reward at $s$ and $f_{\text{out}}(s)$ is the density of outflow at $s$. A natural measure-theoretic way of writing these equations as one is via Radon-Nikodym derivation: given two measures $\mu, \nu$; if $\mu(X) = 0 \Rightarrow \nu(X) = 0$ for any measurable $X \subset \mathcal{S}$ then $\mu$ is said to dominate $\nu$ and, by Radon-Nikodym Theorem, there exists a measurable function $\varphi \in L^1(\mu)$ such that $\nu(X) = \int_{x \in X} \varphi(x) d\nu(x)$ for all measurable $X \subset \mathcal{S}$. This $\varphi$ is the Radon-Nikodym derivative $\frac{d\nu}{d\mu}$.

If one has a measure $\lambda$ dominating both $R$ and $F_{\text{out}}$ and if $F_{\text{out}}$ dominated $R$ then

$$\mathbb{P}(\text{STOP}|s) \coloneqq \frac{dR}{dF_{\text{out}}}(s) = \frac{dR}{d\lambda}(s) \times \left(\frac{dF_{\text{out}}}{d\lambda}\right)^{-1}.$$

When $\mathcal{S}$ is discrete, we choose $\lambda$ as the counting measure and we recover the graph formula above. When $\mathcal{S}$ is continuous, we choose $\lambda$ as the Lebesgue measure and we recover the second formula.

### A.2.2 EXAMPLE

Consider the $D$-dimensional $W$-width hypergrid case with agent set $I$, see Figure 6. The state space is the finite set $\mathcal{S} = \left(\{1, \cdots, W\}^D\right)^I$, say each agent only observes its own position on the grid so that $\mathcal{O}^{(i)} = \{1, \cdots, W\}^D$. the observation-dependent action space of the $i$-th agent $\mathcal{A}_{o^{(i)}}^{(i)}$ is a subset of $H \coloneqq \{\pm \mathbf{1}_k : 1 \le k \le W\}$ where $\mathbf{1}_k$ is the hot-one array $(0, \cdots, 0, 1, 0, \cdots, 0)$ with a one at the $k$-th coordinate. The set $\mathcal{A}_{o^{(i)}}^{(i)}$ depends on $s$: if $1 < s_k < W$ then $\mathcal{A}_{o^{(i)}}^{(i)} = \{\pm \mathbf{1}_k : 1 \le k \le W\} \cup \{\text{STOP}\}$ but if $s_k = 1$ then $-\mathbf{1}_k \notin \mathcal{A}_{o^{(i)}}^{(i)}$ and similarly if $s_k = W$. The local total action space is then

$$\mathcal{A}_{o^{(i)}}^{(i)} = \{(s, a) \mid 1 \le s_k \le W \text{ and } 1 \le s_k + a_k \le W\} \cup \{\text{STOP}\} \subset \{1, \cdots, W\}^D \times H \cup \{\text{STOP}\}.$$

The local state maps $S^{(i)}$ is $S^{(i)}(o^{(i)}, a) = o^{(i)}$. Since each agent may choose its action freely, for any $s \in \mathcal{S}, \mathcal{A}_s = \prod_{i \in I} \mathcal{A}_{o^{(i)}} / \sim$ however, since $\mathcal{A}_{o^{(i)}}$ depends on $i$ and $s$ then $\mathcal{A} \ne \prod_{i \in I} \mathcal{A}_{o^{(i)}} / \sim$.

The local transition kernel $T^{(i)}$ depends both on the global transition kernel and the policies of all the agents. Two possible choices of transitions depend on whether the agent interacts or not. In the non-interacting case $T_1(s, a) = s + a$. If agents may not occupy the same position then the transition rejects the action if the agent moving would put them in the same position; so $T_2(s, a) = s + a$ if $s + a$ is legal, otherwise $p^{(i)} \circ T_2(s, a) = o^{(i)}$ for some $i$. The simplest $T_2$ is to choose $T_2(s, a) = s$ if $s + a$ is illegal. In this case

$$T_2^{(i)}(o^{(i)}, a^{(i)}) = \mathbb{P}(s + a \text{ is legal}|o^{(i)}, a^{(i)})\delta_{o^{(i)} + a^{(i)}} + \mathbb{P}(s + a \text{ is illegal}|o^{(i)}, a^{(i)})\delta_{o^{(i)}}.$$

Clearly, $\mathbb{P}(s + a \text{ is legal}|o^{(i)}, a^{(i)})$ depends on the policies and positions of all the agents, then so does the local transition kernels $T_2^{(i)}$.

A non-negative measure $\mu$ on $\mathcal{S}$ is any function of the form $\mu(X) = \sum_{x \in X} f(x)$ with $f : \mathcal{S} \to \mathbb{R}_+$ any function. Defining the counting measure $\lambda(X) \coloneqq \sum_{x \in X} 1 = \text{Card}(X)$ we have $\mu = f\lambda$ as measures on $\mathcal{S}$, or equivalently, $\frac{d\mu}{d\lambda} = f$. We may thus translate any reward or probability distribution on such a hypergrid as a measure.

A policy is a Markov kernel $\mathcal{S} \to \mathcal{A}$ such that $S \circ \pi = \text{Identity}$. More concretely, it means we have a function that associates to any state $s$ a probability distribution on $\mathcal{A}$ with support on elements of the form $(s, a)$ with $a \in \mathcal{A}_s$. From the description of measures, such a policy is fully described by a function $q : \mathcal{A} \to \mathbb{R}_+$ such that

$$\forall s \in \mathcal{S}, \sum_{a \in \mathcal{A}_s} q(s, a) = 1.$$

The policy is then $\pi(s) = \sum_{a \in \mathcal{A}_s} q(s, a)\delta_{(s, a)}$.

A GFlowNet on this hypergrid in reward-less notations is given by $(F_{\text{init}}, \pi^*, F_{\text{out}}^*)$. Now, $F_{\text{init}}$ is any measure on $\mathcal{S}$, it may be given by a pre-chosen family of categorical distribution of the finite set

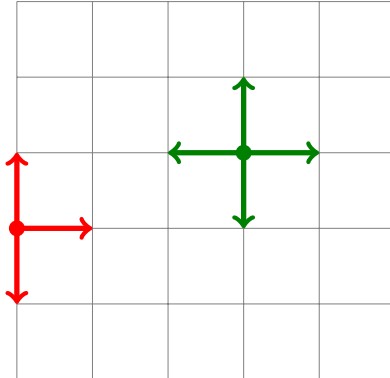

Figure 6: 2 agents on the 2D6W grid with available moves depicted.

$\mathcal{S}$. For big $W, D$ and $I$, since $F_{\text{init}}$ have limited number of parameters, we may choose $F_{\text{init}} = C\delta_{s_1}$ for some $s_1$, and some trainable constant $C$. The star-policy is similar to $\pi$ except that the STOP action is absent:

$$\pi^*(s) = \sum_{a \in \mathcal{A}_s \setminus \text{STOP}} q^*(s, a), \qquad Z(s) := \sum_{a' \in \mathcal{A}_s \setminus \text{STOP}} q(s, a').$$

Finally, $F_{\text{out}}$ is measure and is thus of the form

$$F^*_{\text{out}}(X) := \sum_{x \in X} f^*_{\text{out}}(x)$$

for some function $f^*_{\text{out}} : \mathcal{S} \to \mathbb{R}_+$.

Standard notation GFlowNet is then recovered, given a reward $r : \mathcal{S} \to \mathbb{R}_+$, via:

- $R(X) = \sum_{x \in X} r(x)$;
- $F_{\text{out}}(X) = \sum_{x \in X} f_{\text{out}}(x)$ with $\forall s \in \mathcal{S}, f_{\text{out}}(s) = f^*_{\text{out}}(s) + r(s)$;
- $q(s, a) = \frac{f^*_{\text{out}}(s)}{f_{\text{out}}(s)} q^*(s, a)$ if $a \neq \text{STOP}$ and $q(s, \text{STOP}) = \frac{r(s)}{f_{\text{out}}(s)}$.

A.3 ENVIRONMENT STRUCTURES

We introduce first a hierarchy of single-agent environment structures.

- An action environment is a triplet $(\mathcal{S}, \mathcal{A}, S)$ with $\mathcal{A} \xrightarrow{S} \mathcal{S}$ a measurable map between measurable space is called of state space $\mathcal{S}$, action space $\mathcal{A}$ and State map $S$. We denote $\mathcal{A}_s := \{a \in \mathcal{A} \mid aS = s\}$.
- An interactive environment is a quadruple $(\mathcal{S}, \mathcal{A}, S, T)$ where $(\mathcal{S}, \mathcal{A}, S)$ is an action environment and $T : \mathcal{A} \to \mathcal{S}$ is a quasi-Markov kernel.
- A Game environment is a quintuple $(\mathcal{S}, \mathcal{A}, S, T, R)$ where $(\mathcal{S}, \mathcal{A}, S, T)$ is an interactive environment and $R$ is a finite non-negative non-zero measure on $\mathcal{S}$. We may allow the reward to be stochastic so formally, $R$ is allowed to be random measure instead (Kallenberg et al., 2017).

For multi-agent environment, we have a similar hierarchy:

- A multi-agent action environment is a tuple $(\mathcal{S}, \mathcal{A}, S, \mathcal{O}^{(i)}, \mathcal{A}^{(i)}, S^{(i)}, p^{(i)})_{i \in I}$ with $(\mathcal{S}, \mathcal{A}, S)$ and each $(\mathcal{O}^{(i)}, \mathcal{A}^{(i)}, S^{(i)})$ being mono-agent action environments. Furthermore, we assume $\mathcal{S} = \prod_{i \in I} \mathcal{O}^{(i)}$ and $p^{(i)} : \mathcal{S} \to \mathcal{O}^{(i)}$ are the natural projection maps. Also

$$\forall s \in \mathcal{S}, \quad \mathcal{A}_s \setminus \{\text{STOP}\} = \prod_{i \in I} \left( \mathcal{A}^{(i)}_{p^{(i)}(s)} \setminus \{\text{STOP}\} \right).$$

- A multi-agent interactive environment is a tuple $(\mathcal{S}, \mathcal{A}, S, T, \mathcal{O}^{(i)}, \mathcal{A}^{(i)}, S^{(i)}, p^{(i)})_{i \in I}$ where $(\mathcal{S}, \mathcal{A}, S, \mathcal{O}^{(i)}, \mathcal{A}^{(i)}, S^{(i)}, p^{(i)})_{i \in I}$ is a multi-agent action environment and $(\mathcal{S}, \mathcal{A}, S, T)$ is a mono-agent interactive environment.

- A multi-agent game environment is a tuple $(\mathcal{S}, \mathcal{A}, S, T, R, \mathcal{O}^{(i)}, \mathcal{A}^{(i)}, S^{(i)}, p^{(i)})_{i \in I}$ such that $(\mathcal{S}, \mathcal{A}, S, T, \mathcal{O}^{(i)}, \mathcal{A}^{(i)}, S^{(i)}, p^{(i)})_{i \in I}$ is multi-agent interactive environment and $(\mathcal{S}, \mathcal{A}, S, T, R)$ is a mono-agent game environment.

### A.4 GFLOWNET IN A GAME ENVIRONMENT

A generative flow networks may be formally defined on an action environment $(\mathcal{S}, \mathcal{A}, S)$, as a triple $(\pi^*, F^*_{\text{out}}, F_{\text{init}})$ where $\pi^* : \mathcal{S} \to \mathcal{A}$ is a Markov kernel such that $\pi^* S = Id_\mathcal{S}$, $F^*_{\text{out}}$ and $F_{\text{init}}$ are a finite non-negative measures on $\mathcal{S}$. Furthermore, we assume that for all $s \in \mathcal{S}, \pi^*(s \to \text{STOP}_s) = 0$.

On an interactive environment $(\mathcal{S}, \mathcal{A}, S, T)$, given a GFlowNet $(\pi^*, F^*_{\text{out}}, F_{\text{init}})$, we define the on-going flow as $F_{\text{in}} := F^*_{\text{out}} \pi^* T + F_{\text{init}}$ and the GFlowNet induces an virtual reward $\hat{R} := F_{\text{in}} - F^*_{\text{out}}$. Note that the virtual reward is always finite as the star-outflow and the initial flow are both finite and $\pi^*$ and $T$ are Markovian.

**Definition 1 (Weak Flow-Matching Constraint)** *The weak flow-matching constraint is defined as*

$$\hat{R} \geq 0 \tag{11}$$

If the GFlowNet satisfies the weak flow-matching constraint, we may define a virtual GFlowNet policy as

$$\hat{\pi} := \frac{dF^*_{\text{out}}}{dF_{\text{in}}} \pi^* \tag{12}$$

where $\delta_{\text{STOP}}$ is the deterministic Markov kernel sending any $s$ to $\text{STOP}_s$. The virtual action and edge flows are:

$$\hat{F}_{\text{action}} := F_{\text{in}} \otimes \hat{\pi} \in \mathcal{M}^+(\mathcal{S} \times \mathcal{A}); \tag{13}$$

$$\hat{F}_{\text{edge}} := F_{\text{in}} \otimes (\hat{\pi}T) \in \mathcal{M}^+(\mathcal{S} \times \mathcal{S}). \tag{14}$$

In a game environment, a GFlowNet comes with an outgoing flow, a natural policy, a natural action flow and a natural edge flow

$$F_{\text{out}} := F^*_{\text{out}} + R \tag{15}$$

$$\pi := \frac{dF^*_{\text{out}}}{dF_{\text{out}}} \pi^* \tag{16}$$

$$F_{\text{edge}} := F_{\text{out}} \otimes (\pi T) \in \mathcal{M}^+(\mathcal{S} \times \mathcal{S}) \tag{17}$$

$$F_{\text{action}} := F_{\text{out}} \otimes \pi \in \mathcal{M}^+(\mathcal{S} \times \mathcal{A}). \tag{18}$$

By abuse of notation we also write $F_{\text{action}}$ (resp. $\hat{F}_{\text{action}}$) for $F_{\text{out}}\pi$ (resp. $F_{\text{in}}\pi$). and the flow-matching property may be rewritten as follows.

**Definition 2 (Flow-Matching Constraint)** *The flow-matching constraint on a Game environment $(\mathcal{S}, \mathcal{A}, S, T, R)$ is defined as*

$$\hat{R} = \mathbb{E}(R). \tag{19}$$

**Remark 1** *In an interactive environment $(\mathcal{S}, \mathcal{A}, S, T, \mathcal{O}^{(i)}, \mathcal{A}^{(i)}, S^{(i)}, p^{(i)})_{i \in I}$, a GFlowNet satisfying the weak flow-matching constraint satisfies the (strong) flow-matching constraint on the Game environment $(\mathcal{S}, \mathcal{A}, S, T, \hat{R}, \mathcal{O}^{(i)}, \mathcal{A}^{(i)}, S^{(i)}, p^{(i)})_{i \in I}$.*

We may recover part of the GFlowNets $(\pi^*, F^*_{\text{out}}, F_{\text{init}})$ from any of $F_{\text{action}}, \hat{F}_{\text{action}}$ as in general:

$$\pi^*(x \to A) = \frac{dF_{\text{action}}(\cdot \to A \smallsetminus \text{STOP})}{dF_{\text{action}}(\cdot \to \mathcal{A} \smallsetminus \text{STOP})} = \frac{d\hat{F}_{\text{action}}(\cdot \to A \smallsetminus \text{STOP})}{d\hat{F}_{\text{action}}(\cdot \to \mathcal{A} \smallsetminus \text{STOP})} \tag{20}$$

$$R = F_{\text{action}}(\cdot \to \text{STOP}) \qquad \hat{R} = \hat{F}_{\text{action}}(\cdot \to \text{STOP}) \tag{21}$$

$$F_{\text{out}}^* = F_{\text{action}}(\cdot \to \mathcal{A}) - R = \hat{F}_{\text{action}}(\cdot \to \mathcal{A}) - \hat{R} \tag{22}$$

$$F_{\text{init}} = F_{\text{out}}^* T + \hat{R} \tag{23}$$

If the flow-matching constraint is satisfied, then

$$F_{\text{init}} = F_{\text{out}}^* T + R. \tag{24}$$

Before going further, the presence densities.

**Definition 3** *Let* $\mathbb{F} = (\pi^*, F_{\text{out}}, F_{\text{init}})$ *be a GFlowNet in an interactive environment* $(\mathcal{S}, \mathcal{A}, S, T, \mathcal{O}^{(i)}, \mathcal{A}^{(i)}, S^{(i)}, p^{(i)})_{i \in I}$.

*The initial density of* $\mathbb{F}$ *is the probability distribution*

$$\nu_{\mathbb{F}, \text{init}} := \frac{1}{F_{\text{init}}(\mathcal{S})} F_{\text{init}}$$

*The virtual presence density of* $\mathbb{F}$ *is the probability distribution* $\hat{\nu}_{\mathbb{F}}$ *defined by*

$$\hat{\nu}_{\mathbb{F}} \propto \sum_{t=0}^{\infty} \nu_{\mathbb{F}, \text{init}} \hat{\pi}^t.$$

*The anticipated presence density of* $\mathbb{F}$ *is the probability distribution* $\overline{\nu}_{\mathbb{F}}$ *defined by*

$$\overline{\nu}_{\mathbb{F}} := \frac{1}{F_{\text{in}}(\mathcal{S})} F_{\text{in}}.$$

*In a game environment, the presence density of* $\mathbb{F}$ *is the probability distribution* $\nu_{\mathbb{F}}$ *defined by*

$$\nu_{\mathbb{F}} \propto \sum_{t=0}^{\infty} \nu_{\mathbb{F}, \text{init}} \pi^t.$$

**Lemma 1** *Let* $\mathbb{F}$ *be a GFlowNet in an interactive environment satisfying the weak flow-matching constraint. If* $\hat{\nu}_{\mathbb{F}} \gg \overline{\nu}_{\mathbb{F}}$, *then* $\hat{\nu}_{\mathbb{F}} = \overline{\nu}_{\mathbb{F}}$.

**Proof 1** *Let* $(\mathcal{S}, \mathcal{A}, S, T, \mathcal{O}^{(i)}, \mathcal{A}^{(i)}, S^{(i)}, p^{(i)})_{i \in I}$ *be the interactive environment and let* $\mathbb{F} = (\pi^*, F_{\text{out}}, F_{\text{init}})$. *To begin with,* $\mathbb{F}' := (\pi^*, F_{\text{init}}(\mathcal{S})\hat{\nu}_{\mathbb{F}} - \hat{R}, F_{\text{init}})$ *is a GFlowNet satisfying the strong flow-matching constraint for reward* $\hat{R}$, *its edgeflow* $F'_{\text{edge}}$ *may be compared to the edgeflow* $F_{\text{edge}}$ *of* $\mathbb{F}$*: by Proposition 2 of Brunswic et al. (2024), we have* $F_{\text{edge}} \geq F'_{\text{edge}}$, *and the difference* $F_{\text{edge}} - F'_{\text{edge}}$ *is a 0-flow in the sense this same article. Also, the domination hypothesis implies that* $F'_{\text{edge}} \gg F_{\text{edge}} \gg F^0_{\text{edge}} := F_{\text{edge}} - F'_{\text{edge}}$. *Since the edge-policy of* $F_{\text{edge}}$ *is the same as that of* $F'_{\text{edge}}$ *we deduce that it is also the same as* $F^0_{\text{edge}}$. *By the same Proposition 2, we have* $F'_{\text{out}}\pi^t \xrightarrow{t \to +\infty} 0$, *therefore,* $\mu\pi^t \xrightarrow{t \to +\infty} 0$ *for any* $\mu \ll F'_{\text{out}}$. *Again by domination,* $F'_{\text{edge}} \gg F^0_{\text{edge}}$ *we deduce that* $F'_{\text{out}} \gg F^0_{\text{out}}$. *Therefore,* $F^0_{\text{out}}\pi^t \xrightarrow{t \to +\infty} 0$. *Finally, since* $^0$ *is a 0-flow,* $F^0_{\text{out}}\pi = F^0_{\text{out}}$, *we deduce that* $F^0_{\text{out}} = 0$ *and thus* $F_{\text{edge}} = F'_{\text{edge}}$ *ie* $\hat{\nu}_{\mathbb{F}} = \overline{\nu}_{\mathbb{F}}$.

**Remark 2** *As long as the GFlowNets considered are trained using an FM-loss on a training training distribution* $\nu_{\text{state}}$ *extracted from trajectory distributions* $\hat{\nu}_{\mathbb{F}}$ *or* $\nu_{\mathbb{F}}$ *of the GFlowNets themselves, we may assume that* $\hat{\nu}_{\mathbb{F}} \gg \overline{\nu}_{\mathbb{F}}$ *as flows are only evaluated on a distribution dominated by* $\nu_{\mathbb{F}}$. *The singular part with respect to* $\nu_{\mathbb{F}}$ *is irrelevant for training purposes as well as inference purposes. Therefore, we may generally assume that* $\hat{\nu} = \overline{\nu}$

**Remark 3** *The main interest of the virtual reward* $\hat{R}$ *is for cases where the reward is not accessible or expensive to compute. Since a GFlowNet satisfying the weak flow-matching property always satisfies the strong flow-matching property for the reward* $\hat{R}$, *the sampling Theorem usually applies to* $\hat{R}$. *Therefore,* $\hat{R}$ *may be used as a reward during inference instead of the true reward* $R$ *so that we actually sample using the policy* $\hat{\pi}$ *instead of* $\pi$.

A.5  MA-GFLOWNETS IN MULTI-AGENT ENVIRONMENTS (I): PRELIMINARIES

To begin with, let us define a MA-GFlowNet on a multi-agent environment.

**Definition 4** *An MA-GFlowNet on a multi-agent action environment is the data of a global GFlowNet $\mathbb{F}$ on $(\mathcal{S}, \mathcal{A}, S)$ and a collection of local GFlowNets $\mathbb{F}^{(i)}$ on $(\mathcal{O}^{(i)}, \mathcal{A}^{(i)}, S^{(i)})$ for $i \in I$.*

We give ourselves a multi-agent interactive environment $(\mathcal{S}, \mathcal{A}, S, T, \mathcal{O}^{(i)}, \mathcal{A}^{(i)}, S^{(i)}, p^{(i)})$. We wish to clarify the links between local and global GFlowNet.

- A priori, there the local GFlowNets are merely defined on an action environment, they lack both the local transition kernel $T^{(i)}$ and the reward $R^{(i)}$.
- Given a global GFlowNet, we wish to define local GFlowNets.
- Given a family of local GFlowNets, we wish to define a global GFlowNet.

For simplicity sake, for any GFlowNet $\mathbb{F}$ defined on an interactive environment satisfying the weak flow-matching constraint, we set $R = \hat{R}$ and apply remark 2 assume that $\hat{\nu}_{\mathbb{F}} = \overline{\nu}_{\mathbb{F}} = \nu_{\mathbb{F}}$.

**Definition 5** *Let $(\mathcal{S}, \mathcal{A}, S, T, \mathcal{O}^{(i)}, \mathcal{A}^{(i)}, S^{(i)}, p^{(i)})$ be a multi-agent interactive environment and let $\mathbb{F} = (\pi^*, F^*_{\text{out}}, F_{\text{init}})$ be a GFlowNet on $(\mathcal{S}, \mathcal{A})$ satisfying the weak flow-matching constraint. We introduce the following:*

- *the local presence probability distribution $\nu_{\mathbb{F}}^{(i)} := \nu_{\mathbb{F}} p^{(i)}$;*
- *the measure map $o^{(i)} \mapsto \nu_{\mathbb{F}|o^{(i)}}$ as the disintegration of $\nu_{\mathbb{F}}$ by $p^{(i)}$*
- *the Markov kernel $\tilde{\pi}^{(i)} : \mathcal{O}^{(i)} \to \mathcal{A}$ by $\delta_{o^{(i)}} \tilde{\pi}^{(i)} := \nu_{\mathbb{F}|o^{(i)}} \pi$ ;*
- *the Markov kernel $\pi^{(i)} : \mathcal{O}^{(i)} \to \mathcal{A}^{(i)}$ by $\pi^{(i)} = \tilde{\pi}^{(i)} p^{(i)}$;*
- *the Markov kernel $T^{(i)} : \mathcal{A}^{(i)} \to \mathcal{O}^{(i)}$ by $T^{(i)} = S^{(i)} \tilde{\pi}^{(i)} T p^{(i)}$;*

*The situation may be summarized by the following diagram:*

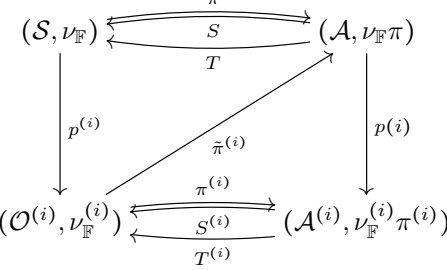

Before going further, we need to check that these definitions are somewhat consistent.

**Lemma 2** *The following diagrams are commutative in the category of probability spaces.*

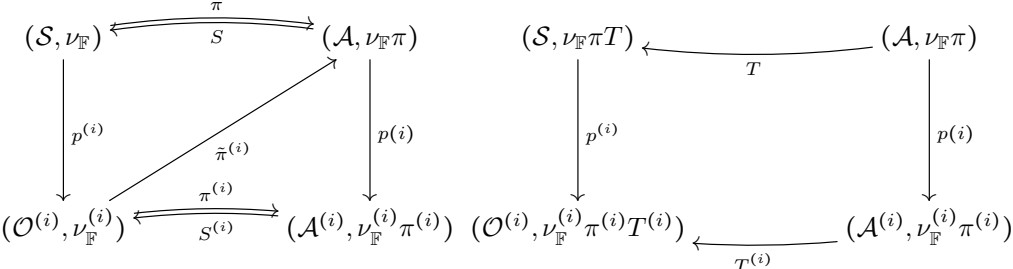

**Proof 2** *For the left diagram, with the definition chosen, we only need to check that $\nu_{\mathbb{F}}^{(i)}\tilde{\pi}^{(i)} = \nu_{\mathbb{F}}\pi$. For all $\varphi \in L^1(\mathcal{A}, \nu_{\mathbb{F}}\pi)$ we have*

$$
\begin{aligned}
\int_{s\in\mathcal{A}} \varphi(a)d(\nu_{\mathbb{F}}\pi)(a) &= \int_{s\in\mathcal{S}} \int_{a\in\mathcal{A}} \varphi(a)d\pi(s,a)d\nu_{\mathbb{F}}(s) \\
&= \int_{o^{(i)}\in\mathcal{O}^{(i)}} \int_{s\in(p^{(i)})^{-1}(o^{(i)})} \int_{a\in\mathcal{A}} \varphi(a)d\pi(s,a)d\nu_{\mathbb{F}|o^{(i)}}(s)d\nu_{\mathbb{F}}^{(i)}(o^{(i)}) \\
&= \int_{o^{(i)}\in\mathcal{O}^{(i)}} \int_{a\in\mathcal{A}} \varphi(a)d\tilde{\pi}^{(i)}(a)d\nu_{\mathbb{F}}^{(i)}(o^{(i)}) \\
&= \int_{a\in\mathcal{A}} \varphi(a)d(\nu_{\mathbb{F}}^{(i)}\tilde{\pi}^{(i)})(a).
\end{aligned}
$$

*For the right diagram, we need to check that $\nu_{\mathbb{F}}\pi p^{(i)} = \nu^{(i)}\pi^{(i)}$ and that $\nu_{\mathbb{F}}\pi T p^{(i)} = \nu_{\mathbb{F}}^{(i)}\pi^{(i)}T^{(i)}$. We already proved the first equality for the left diagram and for the second:*

$$
\nu_{\mathbb{F}}\pi p^{(i)}T^{(i)} := \nu_{\mathbb{F}} \underbrace{\pi p^{(i)}S^{(i)}}_{=p^{(i)}} \tilde{\pi}^{(i)}T p^{(i)} = \underbrace{\nu_{\mathbb{F}}p^{(i)}}_{\nu_{\mathbb{F}}^{(i)}} \tilde{\pi}^{(i)}T p^{(i)} = \nu_{\mathbb{F}}^{(i)}\pi^{(i)}T^{(i)}
$$

We see that from a global GFlowNet, we may build local policies as well as local transition kernels. These policies and transitions are natural in the sense that of local the induced local agent policy an transition are exactly the one wed would have if the observations of the other agents were provided as a random external parameter. The local rewards are then stochastics depending on the state of the global GFlowNet.

### A.6 MA-GFLOWNETS IN MULTI-AGENT ENVIRONMENTS (II): FROM LOCAL TO GLOBAL

We would like to settle construction of global GFlowNet from local ones, key difficulties arise:

- the global distributions induce local ones but the coupling of the local distributions may be non trivial;
- the defining the star-outflow and initial flow requires to find proportionality constants

$$
F_{\mathrm{in}}(\mathcal{O}^{(i)}) \propto \nu_{\mathbb{F}}^{(i)} \qquad F_{\mathrm{init}}^{(i)} \propto \nu_{\mathbb{F}^{(i)},\mathrm{init}};
$$

- The coupling of the local transition kernels $T^{(i)}$ and the global one is in general non-trivial.

We try to solve these issues by looking at the simplest coupling: independent local agents. Recall that $\mathcal{A}_s^* = \prod_{i\in I} \mathcal{A}_s^{(i),*}$ therefore, independent coupling means that $\pi^*(s \to \cdot) = \prod_{i\in I} \pi^{(i),*}(o^{(i)} \to \cdot)$. We may generalize this relation to a coupling of GFlowNets writing $F_{\mathrm{action}}(\prod_{i\in I} O^{(i)} \to \prod_{i\in I} A^{(i)}) = \prod_{i\in I} F_{\mathrm{action}}^{(i)}(O^{(i)} \to A^{(i)})$. We are led to following the definition:

**Definition 6** *Let $(\mathcal{S}, \mathcal{A}, S, T, \mathcal{O}^{(i)}, \mathcal{A}^{(i)}, S^{(i)}, p^{(i)})$ be a multi-agent interactive environment and let $\mathbb{F} = (\pi^*, F_{\mathrm{out}}^*, F_{\mathrm{init}})$ be a global GFlowNet on it satisfying the weak flow-matching constraint. The GFlowNet $\mathbb{F}$ is said to be*

- *star-split if for some local GFlowNets $\mathbb{F}^{(i)}$ and $\forall A^{(i)} \subset \mathcal{A}^{(i)} \smallsetminus \mathrm{STOP}$ we have:*

$$
F_{\mathrm{action}}(\prod_{i\in I} A^{(i)}) = \prod_{i\in I} F_{\mathrm{action}}^{(i)}(A^{(i)}). \tag{25}
$$

- *strongly star-split if for some local GFlowNets $\mathbb{F}^{(i)}$ and $\forall A^{(i)}, B^{(i)} \subset \mathcal{O}^{(i)}$ we have:*

$$
F_{\mathrm{edge}}(\prod_{i\in I} A^{(i)} \to \prod_{i\in I} B^{(i)}) = \prod_{i\in I} F_{\mathrm{edge}}^{(i)}(A^{(i)} \to B^{(i)}). \tag{26}
$$

*The local GFlowNets $\mathbb{F}^{(i)}$ are called the components of the global GFlowNet $\mathbb{F}$.*

However we encounter an additional difficulty: what happens when an agent decides to stop the game ? Indeed, local agents have their own STOP action, we then have at least three behaviors.

1. Unilateral Stop: if any agent decides to stop, the game stops and reward is awarded.

2. Asynchronous Unanimous Stop: if an agent decides to stop, it does not act anymore, waits for the other to leave the game and then reward is awarded only when all agents stopped.

3. Synchronous Unanimous Stop: if an agent decides to stop but some other does not, then the stop action is rejected and the agent plays a non-stopping action.

Similar variations may be considered for how the initialization of agents:

1. Asynchronous Start: the game has a free number of player, agents may enter the game while other are already playing.

2. Synchronous Start: the game has a fixed number of players, and agents all start at the same time.

These 6 possible combinaisons leads to slight variations on the formalization of MA-GFlowNets from local GFlowNets.

## A.7   INITIAL LOCAL-GLOBAL CONSISTENCIES

Let us formalize Asynchronous and Synchronous starts. In synchronous case, the agents are initially distributed according to their own initial distributions and independently. Therefore, $\nu_{\text{init}}$ is a product and

$$F_{\text{init}} \propto \nu_{\text{init}} = \prod_{i \in I} \nu_{\text{init}}^{(i)} \propto \prod_{i \in I} F_{\text{init}}^{(i)}.$$

Also, by strong star-splitting property, $F_{\text{in}}^* = \prod_{i \in I} F_{\text{in}}^{(i),*}$. By $F_{\text{in}} = F_{\text{init}} + F_{\text{in}}^*$ we obtain the definition below.

**Definition 7** *A strongly star-split global GFlowNet is said to have Synchronous start if*

$$F_{\text{in}} = \prod_{i \in I} F_{\text{init}}^{(i)} + \prod_{i \in I} F_{\text{in}}^{(i),*}$$

On the other hand, in the asynchronous case, an incoming agent may "bind" to agent arriving at the same time and other already there hence, the initial flow is a combination of any of the products

$$F_{\text{init}} = \sum \prod_{i \in \{\text{incoming}\}} F_{\text{init}}^{(i)} \prod_{j \in \{\text{already in}\}} F_{\text{init}}^{(j),*} = \prod_{i \in I} (F_{\text{init}}^{(i)} + F_{\text{in}}^{(i),*}) - \prod_{i \in I} F_{\text{in}}^{(i),*}.$$

**Definition 8** *A strongly star-split global GFlowNet is said to have Asynchronous start if*

$$F_{\text{in}} = \prod_{i \in I} (F_{\text{init}}^{(i)} + F_{\text{in}}^{(i),*}).$$

## A.8   TERMINAL LOCAL-GLOBAL CONSISTENCIES

We focus on terminal behaviors 1 and 2 which we formalize as follows. Local-global consistency consists in describing the formal structure linking local environments with global ones. The product structure of the action space is slightly different depending on the terminal behavior. It happens that we may up to formalization, we may cast Asynchronous Unanimous STOP as a particular case of Unilateral STOP local-global consistency. More precisely:

**Definition 9 (Unilateral STOP Local-Global Consistency)** *With the same notations as above, we say that a multi-agent action environment has unilateral STOP if*

$$\mathcal{A}_s := \left( \prod_{i \in I} \mathcal{A}_{o^{(i)}} \right) / \sim \qquad a_1 \sim a_2 \Leftrightarrow \exists i, j \in I, a_1^{(i)} = \text{STOP}^{(i)}, a_2^{(j)} = \text{STOP}^{(j)}. \tag{27}$$

**Definition 10 (Asynchronous Unanimous STOP Local-Global Consistency)** *With the same notations as above, we say that a multi-agent game environment has Asynchronous Unanimous*

*STOP if is has Unilateral STOP and the observation space $\mathcal{O}^{(i)}$ may be decomposed into $\mathcal{O}^{(i)} = \mathcal{O}_{life}^{(i)} \cup \mathcal{O}_{purgatory}^{(i)}$ and for any observation $o^{(i)} \in \mathcal{O}_{life}^{(i)}$ we have some $\tilde{o}^{(i)} \in \mathcal{O}_{purgatory}^{(i)}$ such that :*

$$o^{(i)} \xrightarrow{\hspace{2cm}} \tilde{o}^{(i)} \xrightarrow[\text{STOP}^{(i)}]{R^{(i)}(\tilde{o}^{(i)})} s_f$$
$$\overset{\varepsilon}{\circlearrowright} \qquad \underset{\text{STOP}^{(i)}}{\overset{0}{\longrightarrow}}$$

*where the value on top of arrows are constrained flow values.*

The formal definition of Unilateral STOP is straightforward as any local STOP activates the global STOP so that any combination of local actions that contains at least one STOP is actually a global STOP. The quotient by the equivalence relation formalizes this property. Regarding Asynchronous Unanimous STOP, the chosen formalization allows to store the last observation of an agent while it is put on hold until global STOP. Indeed, a standard action ($\neq$ STOP) is invoked to enter purgatory, the reward is supported on purgatory and as long as all the agent are not in purgatory its value is zero (recall that from the viewpoint of a given agent, $R^{(i)}$ is stochastic but in fact depends on the whole global state). The local STOP action is then never technically called on an "alive" observation, once in purgatory the $\varepsilon$ self-transition is called by default as long as the reward is non zero, hence until all agents are in purgatory. When the reward is activated, the policy at a purgatory state $\tilde{o}^{(i)}$ is then $\frac{d\varepsilon}{d(\varepsilon + R^{(i)})}\delta_{\tilde{o}^{(i)}} + \frac{dR^{(i)}}{d(\varepsilon + R^{(i)})}\delta_{\text{STOP}}$. As $\varepsilon \to 0^+$, the policy becomes equivalent to "if reward then STOP, else WAIT". This behavior is exactly the informal description of Asynchronous Unanimous STOP, the formalization is rather arbitrary and does not limit the applicability as it simply helps deriving formulas more easily.

We now prove Theorem 2.

**Theorem 4** *Let $(\mathcal{S}, \mathcal{A}, S, T, \mathcal{O}^{(i)}, \mathcal{A}^{(i)}, S^{(i)}, p^{(i)})$ be a multi-agent interactive environment. Let $\mathbb{F}^{(i)}$ be non-zero GFlowNets on $(\mathcal{O}^{(i)}, \mathcal{A}^{(i)}, S^{(i)})$ for $i \in I$ satisfying the weak flow-matching constraint, then there exists a transition kernel $\tilde{T}$ and a star-split GFlowNet on $(\mathcal{S}, \mathcal{A}, S, \tilde{T}, \mathcal{O}^{(i)}, \mathcal{A}^{(i)}, S^{(i)}, p^{(i)})$ whose components are the $\mathbb{F}^{(i)}$.*

*Furthermore,*

- *if the multi-agent environment is a game environment with Asynchronous Unanimous STOP and if the global GFlowNet satisfies the strong flow-matching constraint on $\prod_{i \in I} \mathcal{O}_{\text{life}}^{(i)}$ then each local GFlowNet satisfies the strong flow-matching constraint on $\mathcal{O}_{\text{life}}^{(i)}$;*

- *if the multi-agent environment is a game environment with Asynchronous Unanimous STOP and if each local GFlowNets satisfy the strong flow-matching constraint on $\mathcal{O}_{\text{life}}^{(i)}$ then $\hat{R} = \prod_{i \in I} \hat{R}^{(i)}$.*

**Proof 3** *We simply define $\mathbb{F} = (\pi^*, F_{\text{out}}^*, F_{\text{init}})$ by $\pi^*(s) := (\prod_{i \in I} \pi^{(i),*}(o^{(i)}))/\sim$ ie the projection on $\mathcal{A}$ of the policy toward $\prod_{i \in I} \mathcal{A}^{(i)}$, then $F_{\text{out}}^*$ as the product of the measures $F_{\text{out}}^{(i),*}$. Then we define $\tilde{T} = \prod_{i \in I} T^{(i)}$ so that $F_{\text{in}}^*(\prod_{i \in I} A^{(i)}) = \prod_{i \in I} F_{\text{in}}^{(i),*}(A^{(i)})$ and $F_{\text{init}} := \prod_{i \in I}(F_{\text{in}}^{(i),*} + F_{\text{init}}^{(i)}) - \prod_{i \in I} F_{\text{in}}^{(i),*}$ as the product measure of the $F_{\text{init}}^{(i)}$. By construction this GFlowNet is star-split.*

*Assume that $\mathbb{F}$ satisfies the strong flow-matching constraint. It follows that for any $A^{(i)} \subset \mathcal{O}_{\text{life}}^{(i)}$ we have*

$$\prod_{i \in I} F_{\text{in}}^{(i)}(A^{(i)}) = \prod_{i \in I} F_{\text{out}}^{(i)}(A^{(i)}) = \prod_{i \in I} F_{\text{out}}^{(i),*}(A^{(i)}).$$

*Since, by assumption, all local GFlowNets satisfy the weak flow-matching constraint, all terms in the left-hand side product are bigger than those in the right-hand side product. Equality may only occur if some term is zero on both sides or if for all $i \in I$, $F_{\text{in}}^{(i)} = F_{\text{out}}^{(i)}$. Since we assume that the $F_{\text{out}}^{(i),*} \neq 0$ we may take all the $A^{(i)} = \mathcal{O}_{\text{life}}^{(i)}$ except one to ensure we are in the later case. We conclude that the strong flow-matching constraint is satisfied for all local GFlowNets on $\mathcal{O}_{\text{life}}^{(i)}$.*

*If the strong flow-matching constraint is satisfied on $\mathcal{O}_{\text{life}}^{(i)}$, then $\hat{R}^{(i)} = R^{(i)} = 0$ on $\mathcal{O}_{\text{life}}^{(i)}$. By construction, $F_{\text{out}}^{(i),*} = F_{\text{init}}^{(i),*} = 0$ on $\mathcal{O}_{\text{purgatory}}^{(i)}$. Therefore, on purgatory, we have*

$$\hat{R} = F_{\text{in}} - F_{\text{out}} = F_{\text{in}}^* - F_{\text{out}}^* = \prod_{i \in I} F_{\text{in}}^{(i),*} - \prod_{i \in I} F_{\text{out}}^{(i),*} = \prod_{i \in I} F_{\text{in}}^{(i),*} = \prod_{i \in I} \hat{R}^{(i)}.$$

## B  ALGORITHMS

Algorithm 3 shows the training phase of the independent flow network (IFN). In the each round of IFN, the agents first sample trajectories with policy

$$o_t^{(i)} = p_i(s_t^{(i)}) \text{ and } \pi^{(i)}(o_t^{(i)} \to a_t^{(i)}), \quad i \in I \tag{28}$$

with $a_t = (a_t^{(i)} : i \in I)$ and $s_{t+1} = T(s_t, a_t)$. Then we train the sampling policy by minimizing the FM loss $\mathcal{L}_{\text{FM}}^{\text{stable}}(\mathbb{F}^{(i),\theta})$ for $i \in I$.

---

**Algorithm 3** Independent Flow Network Training Algorithm for MA-GFlowNets

**Input:** Number of agents $N$, A multi-agent environment $(\mathcal{S}, \mathcal{A}, \mathcal{O}^{(i)}, \mathcal{A}^{(i)}, p_i, S, T, R)$.
**Input:** Local GFlowNets $(\pi^{(i),*}, F_{\text{out}}^{(i),*}, F_{\text{init}}^{(i)})_{i \in I}$ parameterized by $\theta$.
   **while** not converged **do**
      Sample and add trajectories $(s_t)_{t \geq 0} \in \mathcal{T}$ to replay buffer with policy according to equation 28
      Generate training distribution of observations $\nu_{\text{state}}^{(i)}$ for $i \in I$ from train buffer
      Apply minimization step of FM-loss $\mathcal{L}_{\text{FM}}^{\text{stable}}(F_{\text{action}}^{(i),\theta}, R^{(i)})$ for $i \in I$.
   **end while**

---

Algorithm 4 shows the training phase of Conditioned Joint Flow Network (CJFN). In the each round of CJFN, we first sample sample trajectories with policy

$$o_t^{(i)} = p_i(s_t^{(i)}) \text{ and } \pi_\omega^{(i)}(o_t^{(i)} \to a_t^{(i)}), \quad i \in I \tag{29}$$

with $a_t = (a_t^{(i)} : i \in I)$ and $s_{t+1} = T(s_t, a_t)$. Then we train the sampling policy by minimizing the FM loss $\mathbb{E}_\omega \mathcal{L}_{\text{FM}}^{\text{stable}}(F_{\text{action}}^{\theta,\text{joint}}(\cdot; \omega), R)$.

---

**Algorithm 4** Conditioned Joint Flow Network Training Algorithm for MA-GFlowNets

**Input:** Number of agents $N$, A multi-agent environment $(\mathcal{S}, \mathcal{A}, \mathcal{O}^{(i)}, \mathcal{A}^{(i)}, p_i, S, T, R)$.
**Input:** Simple Random distribution $(\Omega, \mathbb{P})$
**Input:** Local GFlowNets $(\pi^{(i),*}, F_{\text{out}}^{(i),*}, F_{\text{init}}^{(i)})_{i \in I}$ parameterized by $\theta$ and $\omega \in \Omega$.
   **while** not converged **do**
      Sample $\omega_1, \cdots, \omega_b \sim \mathbb{P}$ and then trajectories $(s_t^\omega)_{t \geq 0} \in \mathcal{T}$ to replay buffer with policy according to equation 29 for $\omega \in \{\omega_1, \cdots, \omega_b\}$
      Generate training distribution of states/omega $\nu_{\text{state}}^\Omega$ from the train buffer
      Apply minimization step of the FM loss $\mathbb{E}_\omega \mathcal{L}_{\text{FM}}^{\text{stable}}(\mathbb{F}^{\theta,\text{joint}}(\cdot; \omega))$ under the constraint of Weak flow-matching.
   **end while**

---

## C  DISCUSSION: RELATIONSHIP WITH MARL

Interestingly, there are similar independent execution algorithms in the multi-agent reinforcement learning scheme. Therefore, in this subsection, we discuss the relationship between flow conservation networks and multi-agent RL. The value decomposition approach has been widely used in multi-agent RL based on IGM conditions, such as VDN and QMIX. For a given global state $s$ and joint action $a$, the IGM condition asserts the consistency between joint and local greedy action selections in the joint action-value $Q_{\text{tot}}(s, a)$ and individual action values $[Q_i(o_i, a_i)]_{i=1}^k$:

$$\arg\max_{a \in \mathcal{A}} Q_{\text{tot}}(s, a) = \left( \arg\max_{a_1 \in \mathcal{A}_1} Q_1(o_1, a_1), \cdots, \arg\max_{a_k \in \mathcal{A}_k} Q_k(o_k, a_k) \right), \forall s \in \mathcal{S}. \tag{30}$$

**Assumption 1** *For any complete trajectory in an MADAG $\tau = (s_0, ..., s_f)$, we assume that $Q_{tot}^\mu(s_{f-1}, a) = R(s_f)f(s_{f-1})$ with $f(s_n) = \prod_{t=0}^n \frac{1}{\mu(a|s_t)}$.*

**Remark 1** *Although Assumption 1 is a strong assumption that does not always hold in practical environments. Here we only use this assumption for discussion analysis, which does not affect the performance of the proposed algorithms. A scenario where the assumption directly holds is that we sample actions according to a uniform distribution in a tree structure, i.e., $\mu(a|s) = 1/|\mathcal{A}(s)|$. The uniform policy is also used as an assumption in Bengio et al. (2021).*

**Lemma 3** *Suppose Assumption 1 holds and the environment has a tree structure, based on Theorem 2 and IGM conditions we have:*
*1) $Q_{tot}^\mu(s, a) = F(s, a)f(s)$;*
*2) $(\arg\max_{a_i} Q_i(o_i, a_i))_{i=1}^k = (\arg\max_{a_i} F_i(o_i, a_i))_{i=1}^k$.*

Based on Assumption1, we have Lemma 3, which shows the connection between Theorem 2 and the IGM condition. This action-value function equivalence property helps us better understand the multi-flow network algorithms, especially showing the rationality of Theorem 2.

### C.1 Proof of Lemma 3

**Proof 4** *The proof is an extension of that of Proposition 4 in Bengio et al. (2021). For any $(s, a)$ satisfies $s_f = T(s, a)$, we have $Q_{tot}^\mu(s, a) = R(s_f)f(s)$ and $F(s, a) = R(s_f)$. Therefore, we have $Q_{tot}^\mu(s, a) = F(s, a)f(s)$. Then, for each non-final node $s'$, based on the action-value function in terms of the action-value at the next step, we have by induction:*

$$Q_{tot}^\mu(s, a) = \hat{R}(s') + \mu(a|s') \sum_{a' \in \mathcal{A}(s')} Q_{tot}^\mu(s', a'; \hat{R})$$

$$\stackrel{(a)}{=} 0 + \mu(a|s') \sum_{a' \in \mathcal{A}(s')} F(s', a'; R)f(s'), \tag{31}$$

*where $\hat{R}(s')$ is the reward of $Q_{tot}^\mu(s, a)$ and $(a)$ is due to that $\hat{R}(s') = 0$ if $s'$ is not a final state. Since the environment has a tree structure, we have*

$$F(s, a) = \sum_{a' \in \mathcal{A}(s')} F(s', a'), \tag{32}$$

*which yields*

$$Q_{tot}^\mu(s, a) = \mu(a|s')F(s, a)f(s') = \mu(a|s')F(s, a)f(s)\frac{1}{\mu(a|s')} = F(s, a)f(s).$$

*According to Theorem 2, we have $F(s_t, a_t) = \prod_i F_i(o_t^i, a_t^i)$, yielding*

$$\arg\max_a Q_{tot}(s, a) \stackrel{(a)}{=} \arg\max_a \log F(s, a)f(s)$$

$$\stackrel{(b)}{=} \arg\max_a \sum_{i=1}^k \log F_i(o_i, a_i) \tag{33}$$

$$\stackrel{(c)}{=} \left( \arg\max_{a_1 \in \mathcal{A}_i} F_1(o_1, a_1), \cdots, \arg\max_{a_k \in \mathcal{A}_k} F_k(o_k, a_k) \right),$$

*where $(a)$ is based on the fact $F$ and $f(s)$ are positive, $(b)$ is due to Theorem 2. Combining with the IGM condition*

$$\arg\max_{a \in \mathcal{A}} Q_{tot}(s, a) = \left( \arg\max_{a_1 \in \mathcal{A}_1} Q_1(o_1, a_1), \cdots, \arg\max_{a_k \in \mathcal{A}_k} Q_k(o_k, a_k) \right), \forall s \in \mathcal{S}. \tag{34}$$

*we can conclude that*

$$\left( \arg\max_{a_i \in \mathcal{A}_i} F_i(o_i, a_i) \right)_{i=1}^k = \left( \arg\max_{a_1 \in \mathcal{A}_1} Q_i(o_i, a_i) \right)_{i=1}^k.$$

*Then we complete the proof.*

# D    ADDITIONAL EXPERIMENTS

## D.1    HYPER-GRID ENVIRONMENT

### D.1.1    EFFECT OF SAMPLING METHOD:

We consider two different sampling methods of JFN; the first one is to sample trajectories using the flow function $F_i$ of each agent independently, called JFN (IS), and the other one is to combine the policies $\pi_i$ of all agents to obtain a joint policy $\pi$, and then performed centralized sampling, named JFN (CS). As shown in Figure 7, we found that the JFN (CS) method has better performance than JFN (IS) because the error of the policy $\pi$ estimated by the combination method is smaller, and several better samples can be obtained during the training process. However, the JFN (IS) method can achieve decentralized sampling, which is more in line with practical applications.

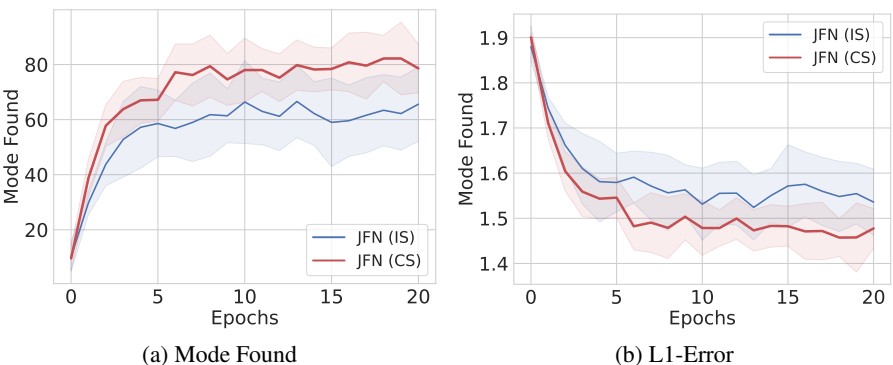

(a) Mode Found          (b) L1-Error

Figure 7: The performance of JFN with different methods.

### D.1.2    EFFECT OF DIFFERENT REWARDS:

We study the effect of different rewards in Figure 8. In particular, we set $R_0 = \{10^{-1}, 10^{-2}, 10^{-4}\}$ for different task challenge. A smaller value of $R_0$ makes the reward function distribution more sparse, which makes policy optimization more difficult Bengio et al. (2021); Riedmiller et al. (2018); Trott et al. (2019). As shown in Figure 8, we found that our proposed method is robust with the cases $R_0 = 10^{-1}$ and $R_0 = 10^{-2}$. When the reward distribution becomes sparse, the performance of the proposed algorithm degrades slightly.

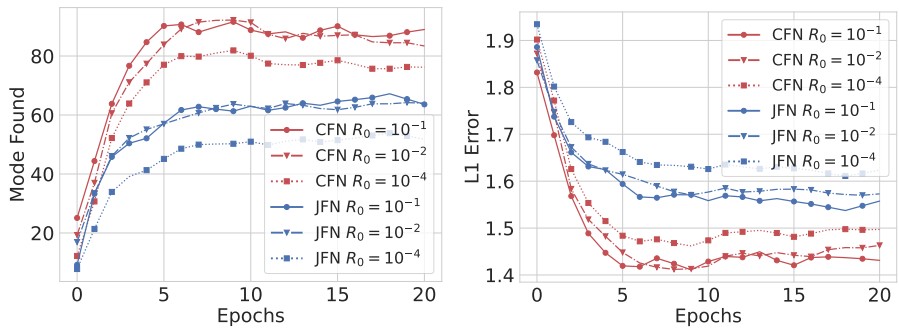

Figure 8: The effect of different reward $R_0$ on different algorithm.

### D.1.3    FLOW MATCH LOSS FUNCTION:

Figure 9 shows the curve of the flow matching loss function with the number of training steps. The loss of our proposed algorithm gradually decreases, ensuring the stability of the learning process.

For some RL algorithms based on the state-action value function estimation, the loss usually oscillates. This may be because RL-based methods use experience replay buffer and the transition data distribution is not stable enough. The method we propose uses an on-policy based optimization method, and the data distribution changes with the current sampling policy, hence the loss function is relatively stable. Then we present the experimental details on the Hyper-Grid environments. We set the same number of training steps for all algorithms for a fair comparison. Moreover, we list the key hyperparameters of the different algorithms in Tables 2-6.

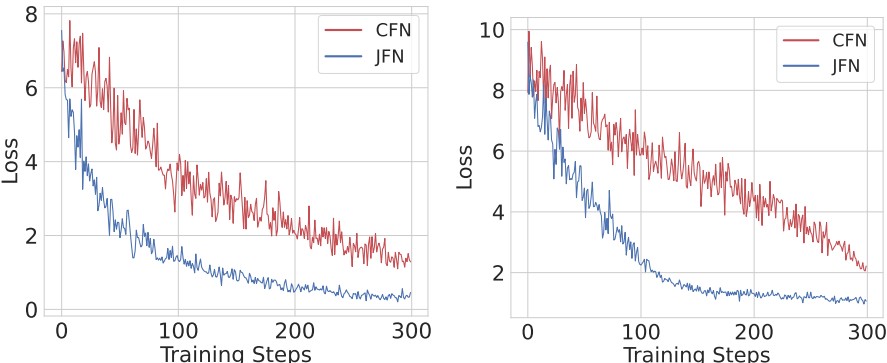

Figure 9: The flow matching loss of different algorithm.

In addition, as shown in Table 1, both the reinforcement learning methods and our proposed method can achieve the highest reward, but the average reward of reinforcement learning is slightly better for all found modes. Our algorithms do not always have higher rewards compared to RL, which is reasonable since the goal of MA-GFlowNets is not to maximize rewards.

| Environment | MAPPO | MASAC | MCMC | CFN | JFN |
|---|---|---|---|---|---|
| Hyper-Grid v1 | 2.0 | 1.84 | 1.78 | 2.0 | 2.0 |
| Hyper-Grid v2 | 1.90 | 1.76 | 1.70 | 1.85 | 1.85 |
| Hyper-Grid v3 | 1.84 | 1.66 | 1.62 | 1.82 | 1.82 |

Table 1: The best reward found using different methods.

## D.2 STARCRAFT

We present a visual analysis based on 3m with three identical entities attacking to win. All comparison experiments adopted PyMARL framework and used default experimental parameters. Figure 10 shows the decision results of different algorithms on the 3m map. It can be found that the proposed algorithm can obtain results under different reward distributions, that is, win at different costs. The costs of other algorithms are often the same, which shows that the proposed algorithm is suitable for scenarios with richer rewards. Figure 11 shows the performance of the different algorithms on 2s3z, which shows a similar conclusion that the algorithm based on GFlowNets may be difficult to get the best yield, but the goal is not to do this, but to fit the distribution better. Moreover, on StarCraft missions, we did not use a clear metric to indicate the diversity of different trajectories, mainly because the status of each entity includes multiple aspects, its movement range, health, opponent observation, etc., which can easily result in different trajectories, but these differences do not indicate a good fit for the reward distribution. As a result, it is not presented in the same way as Hyper-Grid and Simple-Spread. Therefore, we used a visual method to compare the results. The maximized reward-oriented algorithms such as QMIX will improve the reward by reducing the death of entities, while the GFlowNets method can better fit the distribution on the basis of guaranteeing higher rewards.

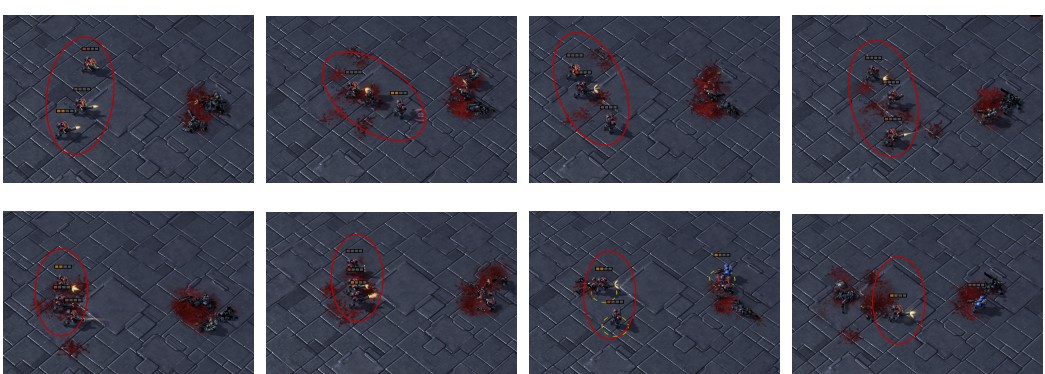

Figure 10: The sample results of different algorithm on 3m map. **Upper**: QMIX, **Bottom**: JFN

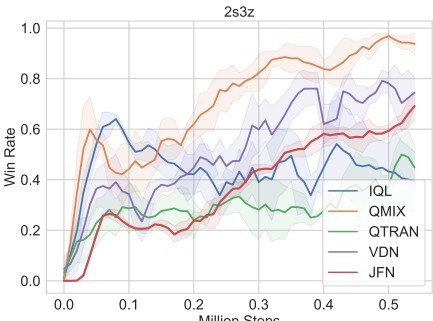

Figure 11: Average rate on 2s3z

### D.3 SPARSE-SIMPLE-SPREAD ENVIRONMENT

In order to verify the performance of the CFN and JFN algorithms more extensively, we also conducted experiments on Simple-Spread in the multi-agent particle environment. We compared two classic Multi-agent RL algorithms, QMIX Rashid et al. (2018) and MAPPO Yu et al. (2022), which have achieved State-of-the-Art performance in the standard simple-spread environment. Since the decision-making problems solved by GFlowNets are usually the setting of discrete state-action space, we modified Simple-Spread to meet the above conditions and named it discrete Sparse-Simple-Spread. Specifically, we set the reward function such that if the agent arrives at or near a landmark, the agent will receive the highest or second-highest reward. And this reward is given to the agent only after each trajectory ends. In addition, we fix the speed of the agent to keep the state space discrete and all agents start from the origin.

We adopt the average return and the number of distinguishable trajectories as performance metrics. When calculating the average return, JFN and CFN select the action with the largest flow for testing. As shown in Figure 12-Left, although the MAPPO and QMIX algorithms converge faster than the JFN, the JFN eventually achieves comparable performance. The performance of JFN is better than that of the CFN algorithm, which also shows that the method of flow decomposition can better learn the flow $F_i$ of each agent. In each test round, we collect 16 trajectories and calculate the number of trajectories, which can be accumulated for comparison. The number of different trajectories found by JFN is 4 times that of MAPPO in Figure 12-Right, which shows that MA-GFlowNets can obtain more diverse results by sampling with the flow function. Moreover, the performance of JFN is not as good as that of the RL algorithm. This is because JFN lacks a guarantee for monotonic policy improvement Schulman et al. (2015; 2017). It pays more attention to exploration and does not fully use the learned policy, resulting in fewer high-return trajectories collected. MAPPO finds more high-return trajectories in Figure 12-Right, but it still struggles to generate more diverse results. In each sampling process, the trajectories found by MAPPO are mostly the same, while JFN does better.

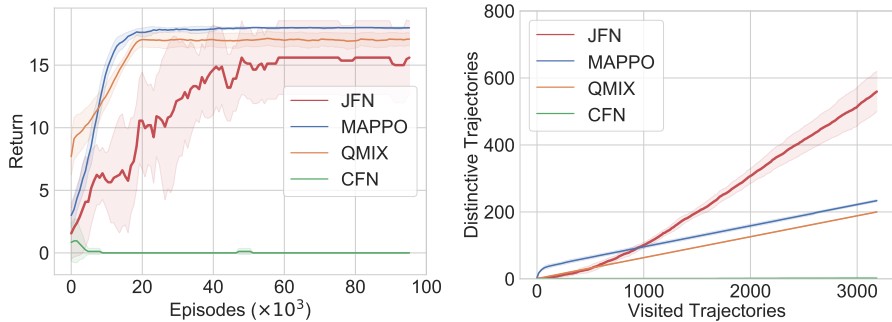

Figure 12: Average return and the number of distinctive trajectories performance of different algorithms on Sparse-Simple-Spread environments.

Table 2: Hyper-parameter of MAPPO under different environments

|  | Hyper-Grid-v1 | Hyper-Grid-v2 | Hyper-Grid-v3 |
| --- | --- | --- | --- |
| Train Steps | 20000 | 20000 | 20000 |
| Agent | 2 | 2 | 3 |
| Grid Dim | 2 | 3 | 3 |
| Grid Size | [8,8] | [8,8] | [8,8] |
| Actor Network Hidden Layers | [256,256] | [256,256] | [256,256] |
| Optimizer | Adam | Adam | Adam |
| Learning Rate | 0.0001 | 0.0001 | 0.0001 |
| Batchsize | 64 | 64 | 64 |
| Discount Factor | 0.99 | 0.99 | 0.99 |
| PPO Entropy | 1e-1 | 1e-1 | 1e-1 |

Table 3: Hyper-parameter of MASAC under different environments

|  | Hyper-Grid-v1 | Hyper-Grid-v2 | Hyper-Grid-v3 |
| --- | --- | --- | --- |
| Train Steps | 20000 | 20000 | 20000 |
| Grid Dim | 2 | 3 | 3 |
| Grid Size | [8,8] | [8,8] | [8,8] |
| Actor Network Hidden Layers | [256,256] | [256,256] | [256,256] |
| Critic Network Hidden Layers | [256,256] | [256,256] | [256,256] |
| Optimizer | Adam | Adam | Adam |
| Learning Rate | 0.0001 | 0.0001 | 0.0001 |
| Batchsize | 64 | 64 | 64 |
| Discount Factor | 0.99 | 0.99 | 0.99 |
| SAC Alpha | 0.98 | 0.98 | 0.98 |
| Target Network Update | 0.001 | 0.001 | 0.001 |

Table 4: Hyper-parameter of JFN under different environments

|  | Hyper-Grid-v1 | Hyper-Grid-v2 | Hyper-Grid-v3 |
| --- | --- | --- | --- |
| Train Steps | 20000 | 20000 | 20000 |
| $R_2$ | 2 | 2 | 2 |
| $R_1$ | 0.5 | 0.5 | 0.5 |
| Grid Dim | 2 | 3 | 3 |
| Grid Size | [8,8] | [8,8] | [8,8] |
| Trajectories per steps | 16 | 16 | 16 |
| Flow Network Hidden Layers | [256,256] | [256,256] | [256,256] |
| Optimizer | Adam | Adam | Adam |
| Learning Rate | 0.0001 | 0.0001 | 0.0001 |
| $\epsilon$ | 0.0005 | 0.0005 | 0.0005 |

Table 5: Hyper-parameter of CJFN under different environments

|  | Hyper-Grid-v1 | Hyper-Grid-v2 | Hyper-Grid-v3 |
| --- | --- | --- | --- |
| Train Steps | 20000 | 20000 | 20000 |
| $R_2$ | 2 | 2 | 2 |
| $R_1$ | 0.5 | 0.5 | 0.5 |
| Grid Dim | 2 | 3 | 3 |
| Grid Size | [8,8] | [8,8] | [8,8] |
| Trajectories per steps | 16 | 16 | 16 |
| Flow Network Hidden Layers | [256,256] | [256,256] | [256,256] |
| Optimizer | Adam | Adam | Adam |
| Learning Rate | 0.0001 | 0.0001 | 0.0001 |
| $\epsilon$ | 0.0005 | 0.0005 | 0.0005 |
| Number of $\omega$ | 4 | 4 | 4 |

Table 6: Hyper-parameter of CFN under different environments

|  | Hyper-Grid-v1 | Hyper-Grid-v2 | Hyper-Grid-v3 |
| --- | --- | --- | --- |
| Train Steps | 20000 | 20000 | 20000 |
| Trajectories per steps | 16 | 16 | 16 |
| $R_2$ | 2 | 2 | 2 |
| $R_1$ | 0.5 | 0.5 | 0.5 |
| Grid Dim | 2 | 3 | 3 |
| Grid Size | [8,8] | [8,8] | [8,8] |
| Flow Network Hidden Layers | [256,256] | [256,256] | [256,256] |
| Optimizer | Adam | Adam | Adam |
| Learning Rate | 0.0001 | 0.0001 | 0.0001 |
| $\epsilon$ | 0.0005 | 0.0005 | 0.0005 |

