# OpenReview forum: "A Theory of Multi-Agent Generative Flow Networks"
_ICLR.cc/2025/Conference — Submitted to ICLR 2025_

### Official Review · Reviewer_eWbR · 2024-11-01

**Soundness:** 1
**Presentation:** 1
**Contribution:** 2
**Rating:** 3
**Confidence:** 5

**Summary:**

This paper proposes a theoretical framework for multi-agent generative flow networks (MA-GFlowNets) and presents four algorithms: Centralized Flow Network (CFN), Independent Flow Network (IFN), Joint Flow Network (JFN), and Conditioned Joint Flow Network (CJFN). The authors introduce a local-global principle based on the principles in MARL that allows training individual GFNs as a unified global GFN. The authors evaluate their approach on Grid and SMAC tasks by comparing with MARL and MCMC approaches.

**Strengths:**

The paper is easy to understand (although some important details are missing as discussed in the next part), and the paper studies an important problem in extending GFlowNets to handle multi-agent tasks.

**Weaknesses:**

1. The paper's novelty claim is questionable. The authors state "However, based on current theoretical results, GFlowNets cannot support multi-agent systems" while ignoring relevant prior work, particularly Li et al. (2023) which already explored multi-agent GFlowNets.
2. The experimental evaluation has several limitations:
- Only uses the simplest 3m StarCraft II environment, and there is little performance improvement.
- Results in Figure 3 show very high L1 errors, which suggests poor learning. Doesn't demonstrate clear advantages over single-agent GFlowNets approaches.
- Little performance improvements over baselines
2. The paper doesn't adequately address the cyclic environment problem. GFlowNets traditionally work best in acyclic environments, but the paper doesn't explain how they handle cycles in StarCraft II scenarios.
3. The motivation for using MA-GFN in the chosen tasks is not well justified. Many of the presented problems could potentially be solved more effectively with single-agent GFlowNets approaches.

**Questions:**

1. How does the proposed approach differ from Li et al. (2023)'s work on multi-agent GFlowNets? Please clarify the novel contributions relative to this prior work.
2. How does the proposed method handle cyclic state transitions in StarCraft II environments, given that GFlowNets traditionally assume acyclic state spaces?
3. The L1 errors shown in Figure 3 are quite high. Could the authors explain why this occurs and how it affects the practical utility of the method? What specific advantages does the MA-GFN approach offer over single-agent GFN solutions for the presented Grid tasks? Could the authors provide experimental comparisons?
4. Why is it evaluated only on the simplest 3m StarCraft II scenario? Have the authors tested your approach on more complex multi-agent scenarios?

---

> ### Author Response · Authors · 2024-11-12
> **Response to Reviewer eWbR**
>
> To begin with, we thank you for the time you took to write this detailed review.
>
> **Response to Question 1:**
> First of all, we generalize the measure GFN framework to the multi-agent framework. In order to reflect the differences of measure GFN framework, we redescribe different algorithms in Li et.al 2023 to serve as the basis for subsequent theoretical analysis. Moreover, we provide the theoretical setting of Global-local principle which justifies the key contribution of the aformentionned work: joint flow based training. Our theory (a) justifies the algorithm with a local-global principle (b) describe shortcomings of this algorithm (c) provide an extension solving these shortcomings via conditionning.
>
> **Response to Question 2:**
> Regarding cycles, we leverage non-acyclic losses defined by Brunswic et al [1]. This prior work provides theoretical account of the acyclic limitation of GFN and how to bypass it via so-called stable losses.
>
> **Response to Question 3:**
> There are two main reasons for the large L1-error. The first is the calculation sampling. In the multi-agent setting, there are a large number of grids that need to be used to calculate the L1-error. For example, the two-agent scene has 4096 grids, but only 16 samples are sampled per round. When calculating the index, we sampled 20 rounds, so the sampling value is much smaller than the number of grids, which will lead to a large L1-error.
> When the number of sampling rounds is increased, the L1-error will be further reduced. When the number of rounds increases to 2000, the normalized L1-error indicator decreases to less than 1.
> But this will increase the additional calculation overhead.
> The second reason is the magnitude of the value. Different from the standard empirical L1 error, we used normalized L1-error, i.e., $\mathbb{E}[|p(x)-\pi(x)|] \times N$, where $p(x)$ is the density of the target $x$, and $N$ is the number of target.
> As the number of final targets with rewards increases, the density of each target will become relatively smaller. In order to visualize the data, an additional scale of the number of grids is multiplied when calculating the L1-error. Moreover, the L1-error is on the order of $10^{-4}$.
> The corresponding comparison has been made on hypergrid. Since the multi-agent problem can also be regarded as a single agent, as the dimension and the number of agents increase, the original GFlowNets often have difficulty solving the above problems.
>
> **Response to Question 4:**  We use 3m scenario to verify the performance of the proposed algorithm. Although this task is the simplest, it is already more complex than the existing research work. In addition, this task is a typical winning-oriented task, which is slightly different from the goal of GFlowNets, but it can still illustrate the fitting ability of the proposed algorithm to the reward distribution.
>
> [1] Leo Maxime Brunswic, Yinchuan Li, Yushun Xu, Shangling Jui, Lizhuang Ma. A Theory of Non-Acyclic Generative Flow Networks. AAAI 2024

---

### Official Review · Reviewer_ZXFS · 2024-11-03

**Soundness:** 2
**Presentation:** 1
**Contribution:** 3
**Rating:** 3
**Confidence:** 3

**Summary:**

The paper presents a theoretical framework focused on adapting GFlowNets to multi-agent setting, building on the previously proposed theory of non-acyclic GFlowNets. Several training algorithms are proposed that can work in centralized and decentralized settings, and experimental evaluation is provided on both synthetic and real-world tasks.

**Strengths:**

This work is one of the first to consider a novel setting of multi-agent GFlowNets, providing extensive theoretical framework and results.

It is known that RL algorithms can be applied to GFlowNet training [1], and this work is among the few [2] that explore the other direction — applying GFlowNets in RL tasks.

References:

[1] Daniil Tiapkin, Nikita Morozov, Alexey Naumov, Dmitry Vetrov. Generative Flow Networks as Entropy-Regularized RL. AISTATS 2024

[2] Leo Maxime Brunswic, Yinchuan Li, Yushun Xu, Shangling Jui, Lizhuang Ma. A Theory of Non-Acyclic Generative Flow Networks. AAAI 2024

**Weaknesses:**

I have two major concerns about this works.

My first major concern is the clarity of the text. For the most part I did not understand the methodological and theoretical results of this paper. The main thing hindering readability is a combination of very heavy mathematical notation with a lack of consistency, clarity and correct order of definitions. Here are some examples:

1) I did not understand what is state map $S$ (line 80) and what is its purpose. It is introduced in Section 2 but never used in the main text of the paper.

2) Why does transition kernel (line 80) only depend on the action, not on state-action pairs? In standard RL and multi-agent RL formulations it depends on both.

3) Can you please explain the equation $\prod_{i \in I} p^{(i)} \circ S \circ \pi=\mathrm{Id}$ (line 94)?

4) The task that multi-agent GFlowNets try to solve is never formally defined. After reading Section 2 one can guess that it is sampling global states with probabilities proportional to the global reward, but the task has to be explicitly stated nevertheless.

5) Local rewards $R^{(i)}$ appear in Section 2 (line 148), but their definition and connection to global reward is given only in the next section.

6) Their definition given in Section 3 is $R^{(i)}\left(o_t^{(i)}\right):=\mathbb{E}\left(R\left(s_t\right) \mid o_t^{(i)}\right)$ (line 189), and they're said to be utilized in the local training loss. From what I understand, this expectation is intractable in the general case, so I do not understand how are they defined in practice. Authors mention that it is possible to use stochastic rewards instead, but as far as I am aware, GFlowNet theory introduced in previous works, upon which this paper builds, does not support stochastic rewards.

7) On line 241, authors mention: "At this stage, the relations between the global/joint/local flow-matching constraints are unclear, and furthermore, the induced policy of the local GFlowNets still depends on the yet undefined local rewards." In my humble opinion, if any novel definition/theorem/algorithm depends on some object, the object has to be previously introduced and properly defined in all cases.

I believe that this paper could greatly benefit from using simplified notation in the main text (while the full set of definitions can be introduced and used in appendix), as well as major revision of Sections 2 and 3 to ensure that the problem itself and all objects we work with are properly defined and explained to the reader in proper order.

My second concern is related to the presentation of experimental results. The abstract states: "Experimental results demonstrate the superiority of the proposed framework compared to reinforcement learning and MCMC-based methods." Conclusion also has a similar statement (line 470). While on toy synthetic hypergrid environment the proposed methods do show significant improvement over baselines, the results on a more interesting StarCraft task do not support this claim. The proposed JFN algorithm falls behind 3 out of 4 baselines and performs similarly to the remaining one (which is Independent Q-Learning, one of the simplest existing algorithms in multi-agent RL). I understand that the main contributions of this paper are theoretical and methodological, but neverhtless I suggest correcting the statements to faithfully reflect the presented results. I also understand that such metric as win rate may not favor GFlowNets compared to RL approaches, but then I would also suggest presenting some other quantitative metric to demonstrate the utility of the proposed approach in this task, e.g. some measure of diversity.

**Questions:**

0) See Weaknesses.

1) Can you please give more detail on how the proposed framework and algorithms differ from the ones presented in [3]?

References:

[3] Shuang Luo, Yinchuan Li, Shunyu Liu, Xu Zhang, Yunfeng Shao, Chao Wu. Multi-Agent Continuous Control with Generative Flow Networks. Neural Networks, Volume 174, 2024

---

> ### Author Response · Authors · 2024-11-12
> **Response to Reviewer ZXFS**
>
> To begin with, we thank you for the time you took to write this detailed review.
>
> **General Response:**
> We are sorry for the misunderstanding caused by the motations in this paper. In order to solve this problem, we first modified the multi-agent setting and GFlowNets from the perspective of measure theory, and explained the misunderstood notations in detail. Then we add a section in the appendix, called **An Introduction for Notations**, which illustrates the necessity of this definition, in the sense that it improves the generality of notations. The structure of Hyper-grid is given to explain the definition of symbols such as policy and flow function.
>
> *Notation Motivation:*
> To begin with, our motivation to  formalize the action space as a measurable bundle $\mathcal{A} := \{(s,a) | s \in \mathcal{S}, a\in \mathcal{A}_s\}\xrightarrow{S} \mathcal{S}$ is three fold:
>
> 1) The available actions from a state may depend on the state itself: on a grid, the actions available while on the boundary of the grid are certainly more limited than while in the middle. More generally, on a graph, actions are typically formalized by edges $s\xrightarrow{a} s'$ of the graph, the data of an edge contains both the origin $s$ and the destination $s'$. In other words, on graphs, actions are bundled with an origin state.  It is thus natural to consider the actions as bundled with the origin state. When an agent is transiting from a state to another via an action, the state map tells where it comes from while the transition map tells where it is going.
>
> 2) We want our formalism to cover as many cases as possible in a unified way: Graphs, vector spaces with linear group actions or mixture of discrete and continuous state spaces. Measures and measurable spaces provide a nice formalism to capture the quantity of reward on a given set or a probability distribution.
>
>
> 3) We want a well-founded and possibly standardized mathematical formalism. In particular, the policy takes as input a state and returns a distribution of actions. the actions should correspond to the input state. To avoid having a cumbersome notion of policy as a family of distributions $\pi_s$ each on $\mathcal{A} _ s$, we prefer to consider the union of the state-dependent action spaces $\mathcal{A}:= \bigcup_{s\in \mathcal{S}} \mathcal{A}_s$ and define the policy as Markov kernel $\mathcal{S}\rightarrow \mathcal{A}$.  However, we still need to satisfy the constraint that the distribution $\pi(s)$ is supported by $\mathcal A_s$.  Bundles are usual mathemcatical objects formalizing such situations and constraints and are thus well suited for this purpose and the constraint is easily expressed as $S\circ \pi(s) = s, \forall s\in\mathcal{S}$.
> \end{enumerate}
>
> Our synthetic formalism comes with a few drawbacks due to the level of abstraction:
>
> 1) The notation $\pi(s)$ differs from the more common notation $\pi(s, a)$ as the action already contains $s$ implicitly.
>
> 2) We need to use Radon-Nikodym derivative.    At a given state, on a graph, a GFlowNets has a probability to stop $$\mathbb P(STOP | s) = \frac{R(s)}{F _ {out}(s)}.$$
>     On a continuous statespace with reference measure $\lambda$, the stop probability is
>     $$\mathbb P(STOP | s) = \frac{r(s)}{f_{\mathrm{out}}(s)}$$
>     where $r(s)$ is the density of reward at $s$ and $f_{\mathrm{out}}(s)$ is the density of outflow at $s$. A natural measure-theoretic way of writing these equations as one is via Radon-Nikodym derivation: given two measures $\mu,\nu$; if $\mu(X)=0 \Rightarrow \nu(X)=0$ for any measurable $X\subset \mathcal{S}$ then $\mu$ is said to dominate $\nu$ and, by Radon-Nikodym Theorem, there exists a measurable function $\varphi \in L^1(\mu)$ such that $\nu(X)=\int_{x\in X}\varphi(x) d\nu(x)$ for all measurable $X\subset \mathcal{S}$. This $\varphi$ is the Radon-Nikodym derivative $\frac{d\nu}{d\mu}$.
>     If one has a measure $\lambda$ dominating both $R$ and $F _ {out}$ and if $F _ {out}$ dominated $R$ then
>     $$\mathbb P(STOP | s) := \frac{dR}{dF _ {out}}(s) = \frac{dR}{d\lambda}(s) \times \left( \frac{dF _ {out}}{d\lambda}\right)^{-1}.$$
>     When $\mathcal{S}$ is discrete, we choose $\lambda$ as the counting measure and we recover the graph formula above. When $\mathcal{S}$ is continuous, we choose $\lambda$ as the Lebesgue measure and we recover the second formula.

---

> > ### Comment · Reviewer_ZXFS · 2024-11-25
> >
> > I thank the authors for their response. However, my concerns regarding the clarity of the text and the presentation of experimental results still remain, thus I decided to keep my score.

---

> > > ### Author Response · Authors · 2024-11-28
> > >
> > > Thank you very much for the reviewer's response, the response you saw earlier was an incomplete version, we have added the complete version. To be specific, we rewrote the entire notation, then explained the need for such a definition, and gave examples to explain the specific meaning of the different motations. We hope you will re-evaluate these responses and thank you again for your time.

---

> > ### Author Response · Authors · 2024-11-28
> > **Response to Reviewer ZXFS (part 2)**
> >
> > **Response to Comments 1,2,3:**
> > Comments1. 2. and 3. are actually related to the same misunderstanding. The action space is a fiber bundler over the state space it the space of couples (position,action). Why is that? The actions available to an agent may depend on the state it is in (say the agent is on the edge of a grid, the move beyond the grid limit is not possible).  Therefore, to each state $s$ correspond available actions $a$ and $S^{-1}(a)$ the set of such actions. $S$ is simply the projection from $(s,a)$ to $s$.
> > The formalism introduced aims at being general but in practice (and in the whole work), we assume that observations contain the whole information, we may thus identify $\mathcal S$ to $\prod_{i\in I} \mathcal O^{(i)}$.
> > The transition map $T$ takes an element of the Action fiber bundle, ie a couple $(s,a)$. It thus depends on both state and action.
> > Finally, with $Id:\prod_{i\in I}\mathcal O^{(i)}\rightarrow\prod_{i\in I}\mathcal O^{(i)}$ the identity map, the equation $\prod_{i\in I} p^{(i)} \circ S \circ \pi = Id$   means that starting from observation $(o^{(i)})_{i\in I}$ one may apply the combined policy $\pi$ to get an action (more precisely a couple state-action), then forget the action to get a state (via the state map $S$) and then recover the observations via the observation projections. This composition should yield the same observation as those we began with.  Despite being obvious in practice, it is a necessary mathematical assumption.
> >
> > **Response to Comments 4:**
> > Indeed, our target consists in sampling states proportionally to the reward the same way a usual GFN would and the same way the centralized MA-GFN does. We added a Problem Formulation section to clarify this in the core of our paper. We clarify how Theorem 2 combined with Theorem 1 answers our problem formulation.
> >
> > **Response to Comments 5,6:**
> > Indeed, the local rewards are untractable, that's actually a key difficulty of localizing GFNs.  They are only used abstractly and in the independent MA-GFN algorithm. And yes, even though GFN could "in principle" work with stochastic reward (say by targeting the expectancy of the reward instead of the random value), and even though MSE-based FM-loss are minimized on this target, to my knowledge attempts were not successful.  The point of our work is to go beyond that by training the collective of MA-GFN on the deterministic reward by enforcing a FM-property of an abstract global GFN.
> >
> > **Response to the  Second Concern:**
> >  We explain this situation from two aspects. First, the main goal of the MA-GFlowNets method is not to achieve higher reward benefits, but to discuss how to retain the characteristics of GFlowNets in a multi-agent setting, that is, the degree of fit between the sample distribution and the reward distribution. Our verification also illustrates this point. Whether it is Hyper-Grid or 3m scenes, it can sample the area of suboptimal rewards.
> > Secondly, the tasks under starcraft are usually win rate-oriented, which is somewhat different from the goal of MA-GFlowNets. Our experiments show that MA-GFlowNets has the potential to solve large-scale decision-making problems while ensuring diversity.

---

### Official Review · Reviewer_CN8s · 2024-11-04

**Soundness:** 2
**Presentation:** 2
**Contribution:** 2
**Rating:** 5
**Confidence:** 5

**Summary:**

This paper studies the theory of multi-agent generative flow networks for co-operative tasks. The paper proposes four learning patterns: the Centralized Flow Network (CFN), Independent Flow Network (IFN), Joint Flow Network (JFN), and Conditioned Joint Flow Network (CJFN) algorithms. The paper also does experiments on the toy hyper-grid environment and one StarCraft game.

**Strengths:**

Originality: The paper is one of the first to study the extension of Gflownets to multi-agent settings.
Quality：The paper proposes four types generative algorithms, and discuss the difference of these algorithms in terms of the training complexity and the performance.
Significance: Experiments validates the proposed method outperforms MAPPO, MCMC in terms of modes found and L1 error.

**Weaknesses:**

1.For the clarity, I would suggest that the authors choose the original Gflownet formulations. The FM formulations in this paper and the original FM paper are quite different, which is quite hard to follow the main idea of this paper.
2.What's the main challenge that extend the Gflownet to multi-agent settings? For now, there seems no technical difficulty for multi-agent Gflownets.
3.The paper only studies the flow matching objective? does the proposed method applies to other Gflownet learning objectives, such as the detailed balance and the trajectory balance loss?
4.For the experiments, which algorithm is the best?  In the common sense, CFN achieves the best performance.  Also, the L1 error of all algorithms are quite high, i.e., these algorithms can not sample the reward distribution in practice. Why does the paper only present the result of JFN on the StarCraft 3m map?

**Questions:**

See the weakness.

---

> ### Author Response · Authors · 2024-11-12
> **Response to Reviewer CN8s**
>
> **Response to Comment 1:**
> Thank you very much for your advice. One of our key contributions are two fold: *(a)* the generalization of the Measure GFN framework under multi-agent and *(b)* a theoretical account of the Joint Flow Loss introduced in Li et al [3].
>
> In particular, our theoretical setting is not limited to DAG or even to continuous setting with absorbing policy such as in Lahlou et al. [2], cycles are allowed via the use of stable losses. The formulation in this manuscript can be regarded as the extension of Bunswic et al. [3], providing a unified description of the algorithms introduced in \cite{luo2024multi} in a more general setting. This allows us to provide a deeper description of the joint flow algorithm, its shortcoming and ways to solve them via conditional JFN.
>
> Regarding notation choices, our definition of GFlowNets is equivalent to the original formulation as well as that of  [2] and [3]. We do not provide all the details of the equivalency but the appendix of [3] provides an equivalency between edgeflow formulation and measure-policy formulation. We are merely decomposition further to better distinguish the parametrizable part of the GFlowNet FlowInit-starpolicy- from say the reward. We modified the paper to smoothen this transition and provide a justification for our choice: in the multi-agent setting, local rewards and local edgeflow of a local agent depend on other agents. The theoretical frameworks becomes burdened with too many implicits and hidden relations between local GFlowNets. Our formulation allows to explicitly separate what the local agent actually controls from what depends on global, possibly intractable, information.  Moreover, in some settings the reward may not be accessible during inference. The stopping condition based on the reward is then replaced by the virtual reward ie $\hat R:=\mathrm{ReLU}(F_{\text{in}}-F{\text{out}})$. Restricting to the starpolicy ensures the GFlowNets using the true reward or the virtual reward are more easily comparable.
>
> **Response to Comment 2:**
> GFN in the multiagent setting may be realized easily in two contexts: *(a)* if the reward is local, then the independent agent has their own independent policy given by a GFN. *(b)* if the reward is global with small communication costs (small observation encoding) and tractable global transitions.
>
> Centralized algorithm is the formalization of *(b)* while independent is the formalizaiton of *(a)* in our framework. We argue in the paper that the condition for a reasonable centralized algorithm is restrictive and that, in general, the reward is global: the Starcraft 3m task is an example where each marine has its own policy, but the reward depends on the state of all three marines at the end of the sequence. The goal of the JFN is to train local agents with independent GFN policies to fit a global reward.
>
> **Response to Comment 3:**
> The key property of the JFN is the decomposition of the action flow of an abstract global GFN as a product of local action flows. Such a property does allow detailed balance or Trajectory balance objectives. DB and FM loss are very closely related and mostly differ by the implementation choice of the backward policy (FM implements the backward policy by finding parents and computing the forward edge flow for each transition to the current state while DB implements an extra model, the backward policy, either fixed or trainable). Unfortunately, Brunswic et al [3] do not provide stable TB loss suitable for the non-acylic case such as the Starcraft 3m task.
>
> [1] S. Luo, Y. Li, S. Liu, X. Zhang, Y. Shao, and C. Wu, “Multi-agent continuous control with generative flow networks,” Neural Networks,vol. 174, p. 106243, 2024.
> [2] S. Lahlou, T. Deleu, P. Lemos, D. Zhang, A. Volokhova, A. Hern´andez-Garcıa, L. N. Ezzine, Y. Bengio, and N. Malkin, “A theory of
> continuous generative flow networks,” in International Conference on Machine Learning. PMLR, 2023, pp. 18 269–18 300.
> [3] L. Brunswic, Y. Li, Y. Xu, Y. Feng, S. Jui, and L. Ma, “A theory of non-acyclic generative flow networks,” in Proceedings of the AAAI
> Conference on Artificial Intelligence, vol. 38, no. 10, 2024, pp. 11 124–11 131

---

> > ### Author Response · Authors · 2024-11-26
> > **Response to Reviewer CN8s (Part 2)**
> >
> > **Response to Comment 4:**
> > *(1)* In some relatively small experiments, i.e., two agents in HyperGrids, CFN works best.
> > However, the performance of the CJFN algorithm is almost very close to that of the CFN algorithm.
> > Moreover, as the number of agents and grid dimensions increase, CFN becomes difficult to find enough patterns.
> > The JFN series of algorithms become more effective. This is mainly because they adopt the divide-and-conquer idea. Each agent only needs to calculate the probability in its own action space, rather than searching in an exponential space.
> >
> > *(2)* There are two main reasons for the large L1-error. The first is the calculation sampling. In the multi-agent setting, there are a large number of grids that need to be used to calculate the L1-error. For example, the two-agent scene has 4096 grids, but only 16 samples are sampled per round. When calculating the index, we sampled 20 rounds, so the sampling value is much smaller than the number of grids, which will lead to a large L1-error.
> > When the number of sampling rounds is increased, the L1-error will be further reduced. When the number of rounds increases to 2000, the normalized L1-error indicator decreases to less than 1.
> > But this will increase the additional calculation overhead.
> > The second reason is the magnitude of the value. Different from the standard empirical L1 error, we used normalized L1-error, i.e., $\mathbb{E}[|p(x)-\pi(x)|] \times N$, where $p(x)$ is the density of the target $x$, and $N$ is the number of target.
> > As the number of final targets with rewards increases, the density of each target will become relatively smaller. In order to visualize the data, an additional scale of the number of grids is multiplied when calculating the L1-error. The actual L1-error is on the order of $10^{-4}$.
> >
> > *(3)* For the 3m scenario, we use it as an example to illustrate the ability of using the generative flow model as a decision model in large-scale decision-making.

---

### Meta-Review · Area_Chair_uXe1 · 2024-12-21

**Metareview:**

The paper introduces a theoretical framework for Multi-Agent Generative Flow Networks (MA-GFlowNets), extending generative flow networks to collaborative multi-agent settings. It proposes four algorithms: centralized, independent, joint, and conditional joint flow networks, aiming to balance centralized training with decentralized execution. While the approach is innovative and demonstrates promising experimental results, the paper lacks sufficient theoretical grounding and fails to clearly differentiate itself from existing work. Additionally, the experimental validation is limited, as it does not adequately explore generalizability across diverse tasks. These weaknesses, particularly the unclear novelty and limited experimental scope, lead to the recommendation for rejection.

**Additional Comments On Reviewer Discussion:**

During the rebuttal period, reviewers highlighted concerns regarding the theoretical grounding of the Multi-Agent Generative Flow Networks (MA-GFlowNets), particularly the lack of rigorous justification for the proposed algorithms and their applicability to broader settings. They also noted limited experimental validation, as the benchmarks used were not diverse enough to demonstrate the generalizability of the approach. The authors responded with clarifications on their framework and provided additional experimental details but did not introduce new evidence or theoretical insights to sufficiently address these issues. These persistent gaps in theoretical rigor and experimental comprehensiveness were key factors in the final recommendation for rejection.

---

### Decision · Program_Chairs · 2025-01-22

Reject